# Efficient manipulation of gene dosage in human iPSCs using CRISPR/Cas9 nickases

Tao Ye [1,2,3,4], Yangyang Duan [1,2], Hayley W. S. Tsang[1,2], He Xu[1,2], Yuewen Chen[1,3,4], Han Cao[1,2], Yu Chen[1,3,4], Amy K. Y. Fu[1,2,3] & Nancy Y. Ip [1,2,3✉]

The dysregulation of gene dosage due to duplication or haploinsufficiency is a major cause of autosomal dominant diseases such as Alzheimer's disease. However, there is currently no rapid and efficient method for manipulating gene dosage in a human model system such as human induced pluripotent stem cells (iPSCs). Here, we demonstrate a simple and precise method to simultaneously generate iPSC lines with different gene dosages using paired Cas9 nickases. We first generate a Cas9 nickase variant with broader protospacer-adjacent motif specificity to expand the targetability of double-nicking–mediated genome editing. As a proof-of-concept study, we examine the gene dosage effects on an Alzheimer's disease patient-derived iPSC line that carries three copies of *APP* (*amyloid precursor protein*). This method enables the rapid and simultaneous generation of iPSC lines with monoallelic, biallelic, or triallelic knockout of *APP*. The cortical neurons generated from isogenically corrected iPSCs exhibit gene dosage-dependent correction of disease-associated phenotypes of amyloid-beta secretion and Tau hyperphosphorylation. Thus, the rapid generation of iPSCs with different gene dosages using our method described herein can be a useful model system for investigating disease mechanisms and therapeutic development.

[1] Division of Life Science, State Key Laboratory of Molecular Neuroscience, Center for Stem Cell Research, Molecular Neuroscience Center, The Hong Kong University of Science and Technology, Clear Water Bay, Hong Kong, China. [2] Hong Kong Center for Neurodegenerative Diseases, Hong Kong, China. [3] Guangdong Provincial Key Laboratory of Brain Science, Disease and Drug Development, HKUST Shenzhen Research Institute; Shenzhen-Hong Kong Institute of Brain Science, 518057 Shenzhen, Guangdong, China. [4] The Brain Cognition and Brain Disease Institute, Shenzhen Institute of Advanced Technology, Chinese Academy of Sciences; Shenzhen-Hong Kong Institute of Brain Science, 518055 Shenzhen, Guangdong, China. ✉email: boip@ust.hk

The complexity of the human brain arises not only from the genes it expresses but also from the precise regulation of gene dosage and expression levels. Their dysregulation due to duplication or haploinsufficiency is a major cause of autosomal dominant diseases, such as Alzheimer's disease (AD)[1,2]. As the most common type of neurodegenerative dementia, AD is characterized by two pathological hallmarks: extracellular amyloid plaques largely comprising amyloid-beta (Aβ) peptides and intracellular neurofibrillary tangles rich in hyperphosphorylated Tau protein[3]. In AD, the duplication of specific genes, such as *APP* (*amyloid precursor protein*) can cause rare early-onset forms of the disease termed "familial AD"[4]. Meanwhile, heterozygous premature termination codon and noncoding variants in a single allele of certain genes[5,6] (e.g., *ABCA7*) confer a risk of late-onset sporadic AD. Given that numerous promising therapeutics showed efficacy in animal models of AD but failed in clinical trials[7], there is a pressing need to establish relevant model systems, such as human neurons that better mimic the pathophysiology of AD in humans.

Technological developments have enabled the generation and subsequent differentiation of human-induced pluripotent stem cells (iPSCs) into various brain cell types and brain organoids[8]. Thus, iPSC-based models with AD-causing mutations can recapitulate certain phenotypes of familial AD, including increased Aβ production, Tau hyperphosphorylation, endosomal abnormalities, and oxidative stress[9–12]. However, when studying genes and variants associated with sporadic AD that might have fewer risk effects, interindividual genetic heterogeneity can hinder the accurate analysis of disease phenotypes in vitro[13]. Therefore, genome-editing tools, such as the bacterial CRISPR (clustered regularly interspaced short palindromic repeats)/Cas9 system, have great potential to generate isogenic iPSC lines that only differ with respect to a specific gene or mutation of interest but are otherwise genetically identical[14,15]. Although heterozygous knockout more accurately models the partial reduction in gene expression caused by noncoding variants, most sporadic AD genes have not been studied in heterozygous-knockout iPSC-based models in parallel to their homozygous-knockout counterparts (summarized in Supplementary Table 1)[16], partly because there is currently no rapid and efficient method to manipulate gene dosage in iPSCs.

RNA-guided Cas9 nuclease enables sequence-specific, double-strand breaks by specifying a 20-nucleotide targeting sequence within its single-guide RNA (sgRNA) followed by a protospacer adjacent motif (PAM) sequence[17]. For the most commonly used *Streptococcus pyogenes* Cas9 ("Cas9" hereafter), the required PAM sequence is NGG[17]. When targeting the exonic regions of a gene, Cas9-induced double-strand breaks can result in frameshift mutations and gene knockout through DNA repair mediated by non-homologous end-joining[18]. As Cas9-mediated gene knockout appears to be highly efficient and insensitive to gene copy number[19], Cas9 with a potent sgRNA tends to yield complete knockout of target genes, which can facilitate loss-of-function studies[20–23] (summarized in Supplementary Table 2). However, given the considerable off-target effects of Cas9 nuclease[24,25], the use of paired Cas9 nickases (Cas9n) can significantly reduce such off-target effects[26]. As both nickases must be functional to generate two single-strand breaks in close proximity (i.e., 0–20 nucleotides between two sgRNAs)[27], the genome targetability of Cas9n is expected to be much lower than that of an individual Cas9 nuclease, thus necessitating further optimization. Furthermore, the ability of paired nickases to manipulate gene dosage in iPSCs has not been evaluated at the single-cell-derived clone level.

In this study, we expanded the genome-editing targetability of paired nickases by generating a Cas9nVQR variant (D1135V/R1335Q/T1337R) with an NGA PAM. In addition, by using an

AD iPSC line carrying *APP* duplication ("APP cells" hereafter), we used paired Cas9nVQR for the one-step generation of isogenic iPSC lines with monoallelic, biallelic, or triallelic knockout of the target gene; among them was an isogenically corrected line ("Iso cells" hereafter) that had undergone monoallelic *APP* disruption (i.e., carrying two functional *APP* alleles) as an *APP* wild-type revertant. After the iPSCs differentiated into cortical neurons, compared to APP neurons, Iso neurons exhibited genotype-dependent rescue of Aβ production and Tau phosphorylation at Thr231. Furthermore, transcriptomic profiling and functional validation revealed a neuronal apoptotic pathway that is potentially implicated in *APP* dosage-dependent AD pathogenesis. Thus, our study demonstrates an efficient approach for rapidly manipulating gene dosage in iPSCs with paired Cas9n, which will facilitate the study of risk genes associated with human diseases.

## Results

**Footprint-free editing of *APP* copy number by paired Cas9 nickases.** To expand the genome-editing targetability of paired nickases, we generated a Cas9nVQR variant (D1135V/R1335Q/T1337R) with an NGA PAM[28] (Supplementary Fig. 1a–d). Genome-wide coverage analysis demonstrated that Cas9n can target 58.1% of 3,209,286,105 sites in the human reference genome (GRCh38). Meanwhile, the Cas9nVQR variant developed in this study can increase the targetability to 79.8% when used alone and to 90.3% when used together with Cas9n; this expanded genome targetability translates into 695,373,139 and 1,032,629,268 genomic sites, respectively, that are not targetable by Cas9n (Supplementary Fig. 1e, Supplementary Software 1). To examine the ability and flexibility of the Cas9nVQR variant to manipulate gene copy number in iPSCs at the single-cell-derived clone level, we utilized an AD iPSC line carrying *APP* duplication (i.e., APP cells)[9] and targeted *APP* exon 16, which is inaccessible to wild-type Cas9n (Fig. 1a). Accordingly, we designed a pair of sgRNAs—designated sgRNA1 and sgRNA2—that targeted this locus (Fig. 1a). To determine their genome-editing efficiency, we transfected constructs that express Cas9nVQR and corresponding sgRNAs into HEK 293T cells. We utilized the endogenous *Eco*RI site in the target sequence to rapidly screen the genome-editing events. An *Eco*RI-resistant band at 1.1 kb indicated the disruption of an endogenous *Eco*RI site and successful editing at the targeted locus (Fig. 1b). Successful genome editing occurred when both sgRNAs and Cas9nVQR were expressed in the HEK 293T cells, whereas no editing occurred when Cas9nVQR was expressed alone or with only one of the two sgRNAs (Fig. 1b).

To establish the *APP* copy number-corrected isogenic line (i.e., Iso cells), we first transfected APP iPSCs with Cas9nVQR-GFP constructs expressing sgRNA1 and sgRNA2 and isolated GFP-positive cells by cell sorting. Next-generation deep sequencing demonstrated an editing efficiency of 40 ± 3.5% in iPSCs after GFP sorting. We subsequently sorted cells for single-clone expansion and screened them by *Eco*RI digestion assay as described above (Fig. 1c). Out of the 12 screened iPSC clones, four exhibited potential genome editing at the target region (Fig. 1d). We did not detect large deletions in the 5.2-kb PCR amplicons surrounding the edited region in these clones; we subsequently genotyped them by Sanger sequencing and next-generation sequencing (Supplementary Fig. 2a). Accordingly, we generated an Iso line (i.e., a corrected line) with two copies of *APP* and lines with one or zero copies of *APP* (Fig. 1e, Supplementary Fig. 2b). Further array-based comparative genomic hybridization (CGH) assay showed that the edited iPSC lines with monoallelic disruption still carried a 722-kb duplication on chromosome 21 like the parent *APP* duplication iPSC line, whereas the iPSC lines with biallelic or triallelic disruption did

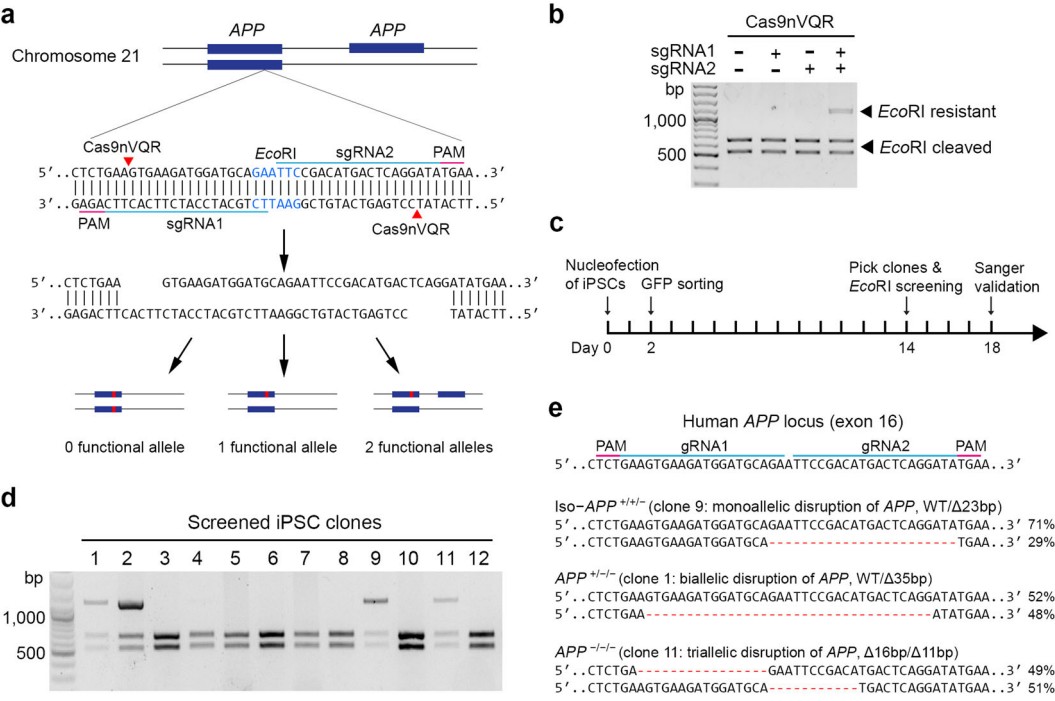

**Fig. 1 Footprint-free gene editing of *APP* copy number in human induced pluripotent stem cells by paired Cas9 nickases. a** Design of a scarless, antibiotic marker-free, gene-editing strategy using a newly generated Cas9 nickase variant, Cas9nVQR (D1135V/R1335Q/T1337R), with paired single-guide RNAs (i.e., sgRNA1 and sgRNA2) that target *APP* exon 16 in neurons with *APP* duplication. Red arrowheads indicate the nickase cleavage site, blue boxes indicate the exon, and the red line within the blue boxes indicates Cas9-induced insertion/deletion (indel) mutations. **b** Genome-editing efficiency of Cas9nVQR with paired sgRNAs in HEK 293T cells. Cas9nVQR with or without corresponding sgRNAs was transfected into HEK 293T cells. The *Eco*RI-resistant band at 1.1 kb indicates successful editing. **c** Schematic diagram of genome editing in a human induced pluripotent stem cell (iPSC) line carrying duplication of the *APP* gene. **d** Screening of gene-edited iPSC clones by *Eco*RI digestion. **e** Specific deletion mutations detected in gene-edited iPSC clones by Sanger sequencing and next-generation deep sequencing. The percentage of deep-sequencing reads for each iPSC clone indicates the number of *APP* copies remaining intact or removed. One isogenic iPSC clone (Iso) had approximately one-third of 23-bp deletion reads and two-thirds of wild-type sequence reads, indicating that one copy of *APP* had been inactivated; therefore, it was used for subsequent experiments. PAM protospacer-adjacent motif.

not (Supplementary Table 3, Supplementary Data 1). These results collectively suggest that paired Cas9nVQR deletes the entire 722-kb duplicate region harboring the *APP* locus in iPSC lines with biallelic or triallelic disruption, which is more than 100-fold larger than that in the previous study (~6 kb)[27]. We used this approach to replicate the efficient generation of the three genotypes simultaneously, demonstrating its reproducibility (Supplementary Fig. 3). Therefore, the double-nicking method flexibly and efficiently manipulated gene copy numbers.

We subsequently examined whether the Iso iPSC lines maintained the pluripotent characteristics of the parental APP iPSC line, specifically pluripotency marker expression and a normal karyotype[9]. Indeed, an Iso iPSC clone stained positive for pluripotent stem cell markers including OCT4, SSEA4, and TRA-1-81 (Supplementary Fig. 4a) and had a normal karyotype upon CGH assay (Supplementary Fig. 4b).

**Impact of *APP* copy correction on amyloid- and Tau-associated pathologies**. We subsequently induced the formation of cortical neurons from corresponding iPSC lines according to the Ngn2 (neurogenin2)-mediated differentiation protocol[15,29]. After 28 days in vitro (DIV), APP, Iso, and nondemented control (NDC) iPSCs efficiently differentiated into forebrain cortical neurons that expressed the neuronal marker MAP2 (Fig. 2a). In addition, more than 90% of MAP2-positive neurons co-expressed the layer II–IV neuronal marker Cux1 (Fig. 2b), suggesting that a homogenous population of upper-layer cortical neurons had been

generated. Moreover, RT-qPCR analysis indicated that the neurons induced from APP, Iso, and NDC iPSCs exhibited similar mRNA expression levels of *NeuN* (a mature neuronal marker), *synaptophysin* (a presynaptic marker), and *PSD-95* (a post-synaptic marker) (Fig. 2c). These results indicate that these neurons had a similar differentiation status.

Neurons derived from APP iPSCs exhibit abnormal Aβ peptide secretion and Tau hyperphosphorylation[9]. Indeed, compared to NDC neurons, APP neurons secreted significantly more $A\beta_{38}$, $A\beta_{40}$, and $A\beta_{42}$ peptides into the medium. Meanwhile, subsequent correction of *APP* gene dosage in Iso iPSCs restored the normal secretion levels of these Aβ peptides (Fig. 2d). Furthermore, in Iso iPSC-derived neurons, the APP protein level decreased close to that in neurons derived from the iPSCs of NDCs, confirming the correction of *APP* copy number in Iso iPSCs (Fig. 2e, g).

We subsequently examined whether *APP* gene copy number affected Tau phosphorylation in Ngn2-induced neurons from APP, Iso, and NDC iPSCs. Tau phosphorylated at Thr231, a component of paired helical filaments Tau, is an early marker of AD pathology correlated with cognitive decline[30]. Compared to NDC neurons, APP neurons exhibited significantly higher Tau phosphorylation at Thr231, total Tau level, and pTau231/total Tau ratio, which is consistent with the previous studies[9,11]. Meanwhile, *APP* copy number correction in Iso neurons reduced the levels of phosphorylated and total Tau (Fig. 2e–g), suggesting that deleting the extra copy of *APP* restored Tau phosphorylation at Thr231, total Tau level, and the pTau231/total Tau ratio.

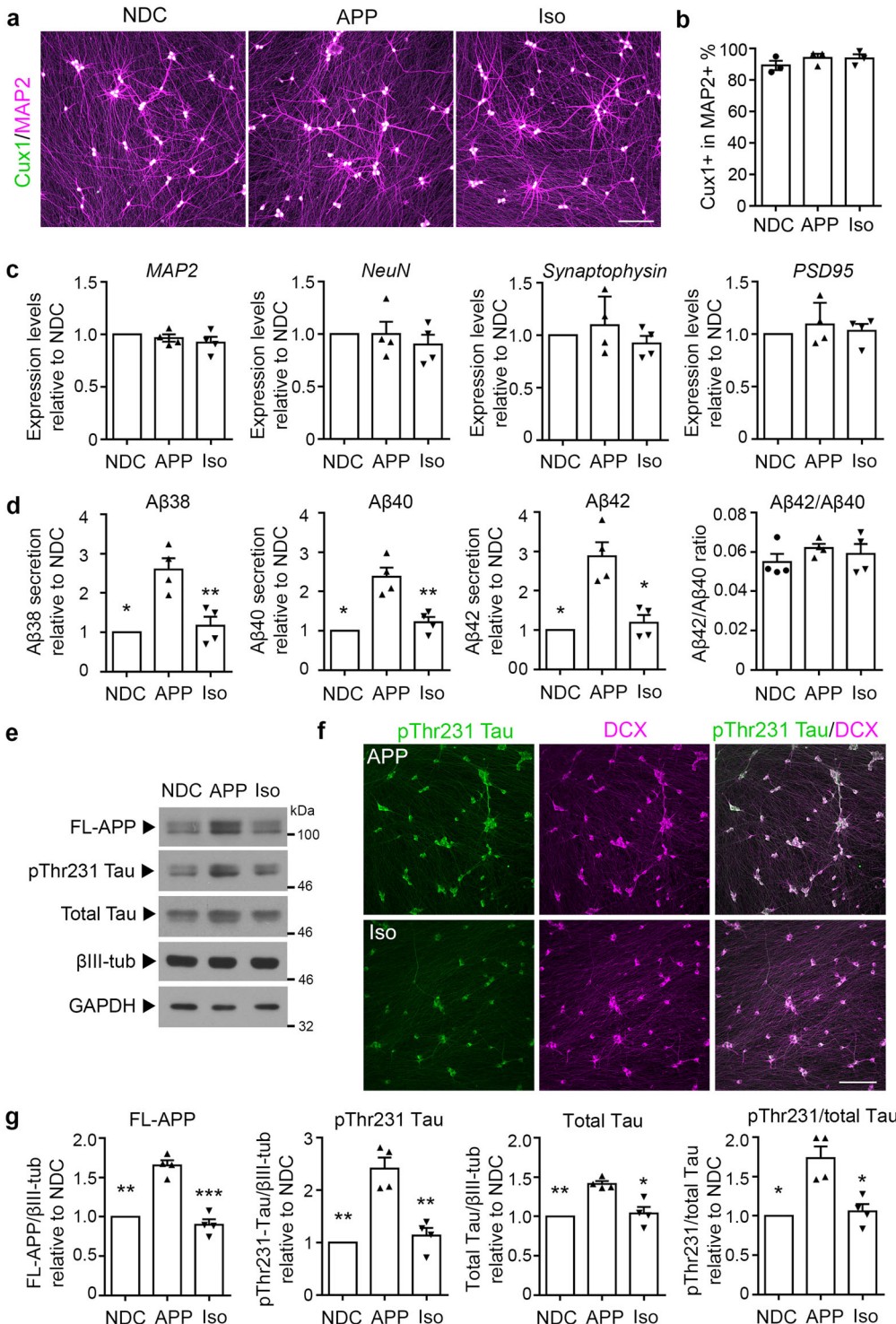

**Fig. 2 CRISPR/Cas9-mediated correction of *APP* copy number alleviates amyloid- and Tau-associated pathologies in induced pluripotent stem cell-derived neurons. a**, **b** Generation of a homogenous population of cortical neurons from nondemented control (NDC), *APP* duplication parent (APP), and *APP* copy number-corrected isogenic (Iso) cell lines. **a** Neurons at 28 days in vitro (DIV) were stained for MAP2 (neuronal marker) and Cux1 (cortical layer II–IV marker). **b** Percentages of Cux1- and MAP2-positive neurons. **c** Neuronal differentiation status as assessed by qRT-PCR for *MAP2*, *NeuN* (mature neuron marker), *Synaptophysin* (presynaptic marker), and *PSD-95* (postsynaptic marker). **d** Levels of secreted amyloid-beta (Aβ)$_{38}$, Aβ$_{40}$, and Aβ$_{42}$ as well as Aβ$_{42}$/Aβ$_{40}$ ratios in conditioned media from NDC, APP, and Iso neurons at 28 DIV. **e** Western blot analysis of full-length APP (FL-APP), phosphorylated Tau at Thr231 (pThr231 Tau), and total Tau in NDC, APP, and Iso neurons at 28 DIV. βIII-tubulin (βIII-tub) and GAPDH served as loading controls for neurons and total protein, respectively. **f** Immunocytochemical analysis of pThr231 Tau and doublecortin (DCX) in APP and Iso neurons at 28 DIV. **g** Quantification of APP/βIII-tub, pThr231 Tau/βIII-tub, total Tau/βIII-tub, and pThr231/total Tau ratios relative to those in NDC neurons. Values are mean ± SEM ($n = 4$ independent biological replicates per cell line; *$p < 0.05$, **$p < 0.01$, or ***$p < 0.001$ vs. APP neurons, Student's *t*-test). Scale bars: 100 μm.

These results are concordant with a recent study showing that the pTau231/total Tau ratio is regulated through mechanisms involving cholesterol metabolism, cholesteryl esters, and proteasome degradation in iPSC-derived neurons with *APP* duplication[31].

**Screening of pathways associated with *APP* gene dosage.** To screen for the molecular pathways associated with *APP* gene dosage, we profiled the transcriptomes of neurons from APP and Iso iPSC lines. Differential gene expression analysis revealed 376 differentially expressed genes (DEGs) between the APP and Iso lines termed "isogenic DEGs." Hierarchical clustering analysis revealed that three batches of APP and Iso neurons clustered together within each respective group (Fig. 3a). Among them, 208 and 168 genes were up and downregulated, respectively, in APP neurons compared to Iso neurons. This suggests that *APP* gene dosage broadly affects gene expression in a human diploid cell system. In parallel, differential gene expression analysis

between the APP and non-isogenic NDC lines yielded 2,158 DEGs termed "non-isogenic DEGs" (including 1050 and 1108 up and downregulated genes, respectively). Venn diagram analysis of these two groups of DEGs revealed 207 common genes (Fig. 3b), which accounted for 55% of the isogenic DEGs (i.e., APP vs. Iso) and less than 10% of non-isogenic DEGs (i.e., APP vs. NDC). Therefore, we analyzed these 207 common genes, which exhibited both *APP*-dependent effects and dysregulation in APP neurons.

Functional annotation of these 207 common DEGs revealed several significantly enriched gene ontology categories including "regulation of transcription" (false discovery rate [FDR] = 4.82E−4), "regulation of response to stimulus" (FDR = 2.88E−2), "regulation of the Wnt signaling pathway" (FDR = 3.84E−2), and "homophilic cell adhesion via plasma membrane adhesion molecules" (FDR = 4.06E−2) (Fig. 3c). Subsequent analysis of the top genes using FDR filters and visualization using volcano plots yielded candidates that might have functional implications in AD pathogenesis (Fig. 3d, e; Table 1). Therefore, we performed RT-qPCR to validate the top four

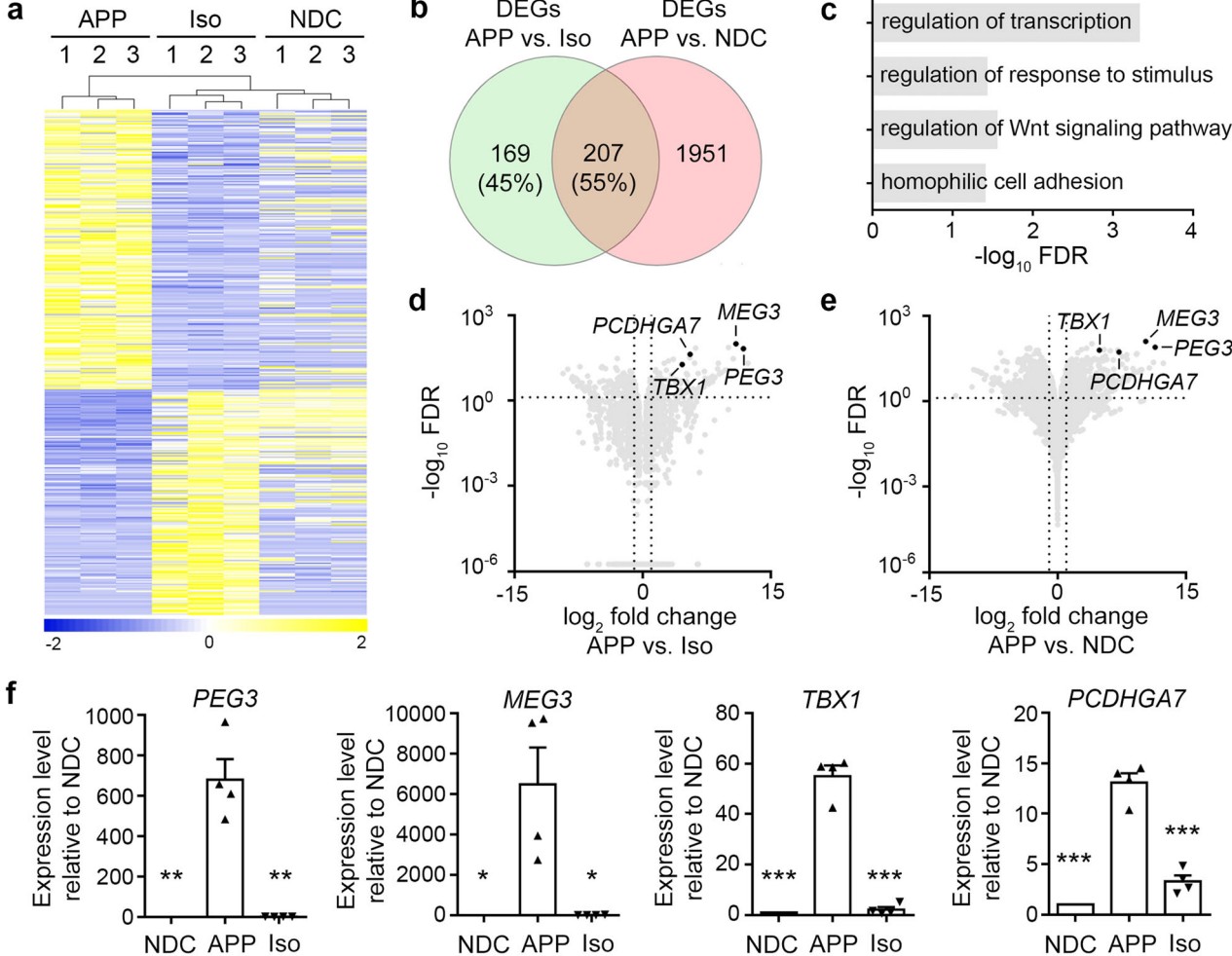

**Fig. 3 Transcriptomic profiling of isogenic induced pluripotent stem cell-derived neurons. a** Hierarchical clustering analysis of the 376 differentially expressed genes (DEGs) in *APP* duplication parent (APP), *APP* copy number-corrected isogenic (Iso), and nondemented control (NDC) neurons. Among them, 208 and 168 genes were up and downregulated, respectively, in APP neurons compared to Iso neurons (false discovery rate [FDR] < 0.05, fold change ≥1.5). **b** Venn diagram illustrating overlap between isogenic DEGs (APP vs. Iso neurons) and non-isogenic DEGs (APP vs. NDC neurons) (FDR < 0.05, fold change ≥1.5). **c** Enriched Gene Ontology (GO) terms for the 207 common DEGs. Volcano plots showing the DEGs between the APP and NDC neurons (**d**) and between the Iso and APP neurons (**e**). Four selected genes upregulated in APP neurons compared to NDC neurons that were the top genes downregulated in copy number-corrected Iso neurons compared to APP neurons. **f** RT-qPCR validation of top DEGs common to both NDC and Iso neurons compared to APP neurons. Values are mean ± SEM (*n* = 4 independent biological replicates per condition; \**p* < 0.05, \*\**p* < 0.01, or \*\*\**p* < 0.001 vs. APP neurons, Student's *t*-test).

**Table 1 Top ten differentially expressed genes common between the APP and NDC neurons and between the Iso and APP neurons.**

| | Gene | Log$_2$ FC | FDR | Function |
|---|---|---|---|---|
| 1 | PEG3 | 11.81 | 1.42E−66 | Bax-mediated cell death |
| 2 | MEG3 | 10.93 | 6.59E−98 | Tumor-suppressor lncRNA |
| 3 | ZNF208 | 10.62 | 6.71E−30 | DNA binding |
| 4 | ZNF736 | 6.2 | 4.61E−70 | DNA binding |
| 5 | PCDHGA7 | 5.56 | 4.16E−43 | Cell adhesion |
| 6 | TBX1 | 4.64 | 4.50E−19 | DNA binding |
| 7 | CRLF1 | 2.55 | 3.56E−21 | Cytokine receptor-like factor |
| 8 | MAPK8 | 1.49 | 2.67E−11 | JNK signaling, Tau phosphorylation |
| 9 | CNTNAP4 | −1.21 | 1.75E−6 | Cell adhesion |
| 10 | SASH1 | −1.27 | 1.06E−6 | TLR4 signaling |

APP APP duplication parent line, FC fold change, FDR false discovery rate, Iso APP copy number-corrected isogenic line, NDC nondemented control line.

DEGs between the APP and NDC lines as well as between the APP and Iso lines (Fig. 3f).

**Inhibition of PEG3–Bax rescues neuronal cell death with APP duplication.** To investigate the possible biological relevance of the identified DEGs, we focused on PEG3 (paternally expressed gene 3), a gene implicated in p53-mediated cell death[32,33]. We first performed western blot analysis to validate that the changes in PEG3 mRNA levels observed in iPSC-derived neurons paralleled the changes in protein levels. PEG3 protein level was significantly higher in APP neurons than NDC neurons but reverted to the basal level upon APP copy number correction (Fig. 4a, b). Given that PEG3 is implicated in the DNA damage-induced neuronal apoptotic pathway[34], we investigated neuronal death in both APP and Iso iPSC-derived neurons. Genetic correction reduced apoptosis as indicated by the number of cleaved caspase 3-positive apoptotic cells in APP and Iso iPSC-derived neurons at 28 DIV (Fig. 4c, d).

Next, we elucidated which downstream effector(s) mediates the action of PEG3 during neuronal death in APP iPSC-derived neurons. Given that PEG3 promotes apoptosis by inducing Bax translocation from the cytosol to the mitochondria[32,34], we examined if inhibition of Bax translocation alleviates neuronal death in APP iPSC-derived neurons. Bax, a pro-apoptotic member of the Bcl-2 protein family, exerts its function in mitochondria-dependent apoptosis[35], while its mitochondrial translocation can be inhibited by BIP-V5 (Bax inhibitor peptide V5), a cell-permeable synthetic peptide inhibitor of Bax[36]. Bax translocation induces the release of cytochrome C in apoptosis, which can be inhibited by CRI (cytochrome C release inhibitor; a Bax channel blocker)[37]. BIP-V5 or CRI administration significantly reduced the percentage of cleaved caspase 3-positive or propidium iodide-positive neurons compared to the vehicle control-treated APP iPSC-derived neurons (Fig. 4e, f; Supplementary Fig. 5), suggesting that inhibiting either Bax translocation or cytochrome C release can rescue apoptosis in these neurons. These results collectively reveal that APP duplication induced PEG3 upregulation and BAX-mediated neuronal apoptosis in a human iPSC model of familial AD carrying APP duplication.

## Discussion

This study demonstrates that the copy number of a given gene can be efficiently manipulated by paired Cas9 nickases. Correcting the copy number of APP rescues several pathological features of AD, including the production of Aβ and hyperphosphorylated Tau. Our transcriptomic analysis reveals key dysregulated pathways in neuronal death and cell stress response among others. Specifically, we show that APP duplication induces PEG3 upregulation and BAX-mediated neuronal apoptosis. Furthermore, inhibition of Bax rescues the neuronal loss in APP-overexpressing, iPSC-derived neurons, suggesting that inhibiting Bax can be explored as a potential therapeutic approach for AD associated with increased APP expression.

The CRISPR/Cas9-mediated manipulation of gene copy number holds great promise for modeling genetic disorders caused by gene duplication or haploinsufficiency, such as AD[4,38]. Genome-wide association studies have identified thousands of risk variants associated with AD; most of these variants reside in noncoding regions surrounding ~30 risk genes[39–41] and have been suggested to modify disease risk by affecting the expression of their respective target genes[42,43]. Although it is impossible to generate an iPSC line for each variant, it is both feasible and of great interest to manipulate the gene dosage by inducing heterozygous and homozygous knockout of these risk genes in iPSCs in order to mimic variant effects, elucidate disease mechanisms, or create relevant models for therapeutic development. Despite the enhanced specificity of Cas9n, the rapid, flexible, and efficient manipulation of gene dosage using Cas9n in human iPSCs have not been demonstrated. One potential obstacle is the restricted targetability of Cas9n. Therefore, we generated a Cas9nVQR variant with significantly expanded targetability, which can potentially edit 90.3% of the 3,209,286,105 sites in the human reference genome (GRCh38); this represents 1,032,629,268 additional sites that are not targetable by Cas9n. Furthermore, while Cas9n exhibits enhanced specificity, it is assumed to be less efficient for genome editing than Cas9 nuclease. In this study, we readily generated integration-free isogenic clones with different gene dosages of APP from the parent iPSCs by screening fewer than 20 clones within 1 month. Considering that there are ~30 uncharacterized sporadic AD risk genes (Supplementary Table 1), it would be possible to simultaneously generate heterozygous- and homozygous-knockout iPSC lines within a reasonable timeframe (e.g., ten genes in 1 month). Hence, our protocol could enable the generation of a large panel of isogenic iPSC lines that are haploinsufficient in the ever-growing number of known disease risk genes, which could be used to uncover shared and distinct pathophysiological mechanisms.

There are several differentiation protocols for generating cortical excitatory neurons in the literature. In this study, we utilized the Ngn2-mediated differentiation protocol to generate cortical neurons with homogenous differentiation status and neuronal identity (i.e., Cux1$^+$ superficial layer neurons)[29]. In contrast, other protocols utilize a chemical-based, dual-SMAD inhibition method[44], which recapitulates the temporal development of the

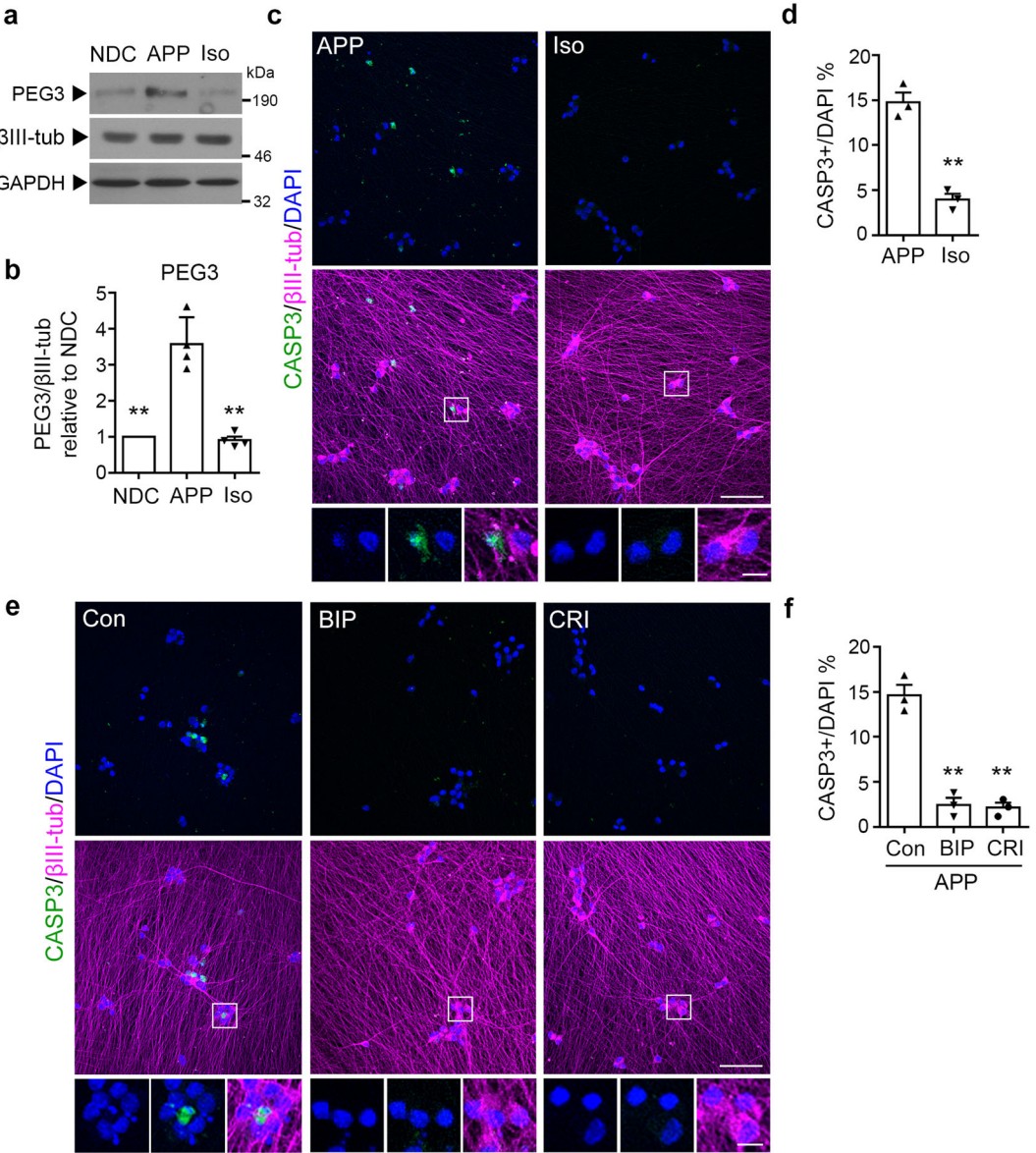

**Fig. 4 PEG3 dysregulation and neuronal death are rescued in gene-corrected isogenic neurons. a** Western blot analysis of PEG3 protein in nondemented control (NDC), *APP* duplication parent (APP), and *APP* copy number-corrected isogenic (Iso) neurons. **b** Quantification of PEG3 expression levels normalized to that of βIII-tubulin (βIII-tub) and relative to those in the NDC neurons. **c** Immunofluorescence images showing CASP3 (cleaved caspase 3)-positive apoptotic cells with pyknotic condensed nuclei in APP and Iso neurons at 28 days in vitro. Scale bars: 50 and 10 μm. **d** Percentage of CASP3-positive apoptotic cells in all neurons. The proportion of apoptotic cells was lower in Iso neurons than that in APP neurons. **b, d** Values are mean ± SEM ($n = 3$ independent biological replicates per line; **$P < 0.01$ vs. APP line; Student's $t$-test). **e** Immunofluorescence images showing CASP3-positive cells in APP neurons at 28 days in vitro treated with vehicle control (Con), 100 μM BIP-V5 (Bax inhibitor peptide V5), or 5 μM CRI (cytochrome C release inhibitor) for 7 days. Scale bars: 50 and 10 μm. **f** Percentage of CASP3-positive apoptotic cells among all neurons. The proportion of apoptotic cells was lower in BIP-V5- or CRI-treated neurons than that in Con neurons. Values are mean ± SEM ($n = 3$ independent biological replicates per line; **$p < 0.01$ vs. Con neurons; Student's $t$-test).

cerebral cortex and produces mixed populations of deep- and upper-layer neurons as well as astrocytes[45]. Such a complex cell type composition might affect results regarding specific AD phenotypes. Thus, our study demonstrates the utility of using an iPSC-derived, single-neuron population to examine *APP* duplication-specific effects on AD pathogenesis.

While brain atrophy caused by neuronal loss is a prominent pathological feature of AD[46], whether and how familial AD-associated *APP* duplication leads to neuronal loss remains poorly understood. Transcriptomic analysis revealed dysregulation in apoptosis-associated genes in APP neurons compared to Iso

neurons. Among them, PEG3 is a tumor-suppressor protein involved in the DNA damage-induced apoptotic pathway via Bax translocation[32,34]. Accordingly, APP neurons exhibited dysregulated PEG3 expression and elevated apoptosis, which were rescued by correcting the extra copy of *APP*. Furthermore, specific inhibition of Bax translocation or cytochrome C release rescued the elevated cell death of APP neurons. These results suggest that *APP* duplication contributes to the increased susceptibility neurons to death via a PEG3–Bax cytochrome C pathway. Together with reports on the neuroprotective role of the inhibition of Bax translocation or cytochrome C release in ischemia[47] and

Huntington's disease[48], our findings suggest that inhibiting Bax can be exploited as a potential therapeutic approach to alleviating neuronal loss in AD.

In summary, by using *APP* as a proof-of-concept, we developed a Cas9nVQR-based genome-editing method that can efficiently manipulate the copy number of specific risk genes with expanded targetability and no footprint. Our study also demonstrates that *APP* regulates Tau expression and hyperphosphorylation and contributes to PEG3 upregulation and BAX-mediated neuronal apoptosis in an isogenic background. Thus, our method can rapidly and simultaneously generate heterozygous and homozygous gene knockout in iPSCs and can be utilized as a model system to mimic gene dosage effects, elucidate disease mechanisms, and facilitate therapeutic development.

## Methods

**Plasmids and antibodies**. The sgRNA sequences for *APP* exon 16 were as follows: sgRNA1: 5′-TCTGCATCCAGATTCACTTC-3′, sgRNA2: 5′-TTCCGACATGAC TCAGGATA-3′. We generated the Cas9nVQR (D1135V/R1335Q/T1337R) variant with altered PAM specificity to the NGA sequence by site-directed mutagenesis (QuikChange II XL Site-Directed Mutagenesis Kit, Stratagene) based on the pSpCas9n(BB)-2A-GFP (PX461) plasmid (Addgene #48140). The newly generated plasmid has been deposited in Addgene (accession code: 165033). For the double-nicking method, we inserted sgRNA1 and sgRNA2 into pSpCas9nVQR(BB)-2A-GFP via the *BbsI* site (New England Biolabs). Then, we transformed the ligation product into NEB Stable Competent *E. coli* (New England Biolabs) for molecular cloning. We prepared plasmid DNA using the E.Z.N.A. Plasmid DNA Mini Kit I (Omega Bio-tek) according to the manufacturer's instructions. We prepared large-scale plasmids in endotoxin-free conditions using the Qiagen Plasmid Maxi Kit (Qiagen).

We used the following primary antibodies for immunofluorescence and western blot analysis: mouse anti-Oct-3/4 (C-10; 1:1,000, #SC-5279, Santa Cruz), anti-SSEA4 (1:1,000, #414000, Life Technologies), anti-TRA-1-81 (1:1,000, #MAB4381, Millipore), anti-Cux1 (1:500, #ab54583, Abcam), anti-APP (6E10; 1:2,000, #803016, Biolegend), anti-human phospho-Tau at Thr231 (AT180; 1:1,000, #MN1040, Thermo Fisher Scientific), anti-total Tau (TAU-5; 1:20,000, #AHB0042, Life Technologies), anti-βIII-tubulin (TUJ1; 1:5,000, #MAB1637, Millipore), anti-GAPDH (1:10,000, GAPDH, Thermo Fisher Scientific), rabbit anti-cleaved Caspase 3 (Asp175; 1:1,000, #9661, Cell Signaling), guinea pig anti-DCX (1:2,000, #AB2253, Millipore), and chicken anti-MAP2 (1:5,000, #ab5392, Abcam).

**Cell lines and cell culture**. Dr. Lawrence Goldstein (University of California, San Diego) generously provided one NDC-derived iPSC line (NDC1.1) and one AD patient-derived iPSC line (APP1.1) carrying *APP* gene duplication, both from male donors[9]. The culture of iPSCs was approved by the Committee on Research Practices of the Hong Kong University of Science and Technology. We cultured iPSCs in feeder-free conditions on Geltrex (Thermo Fisher Scientific) in Essential 8 medium (E8, Thermo Fisher Scientific). We monitored cell quality daily and mechanically removed differentiated cells under a light microscope in a biosafety hood. We cultured iPSCs to 80% confluence and passaged them using Versene EDTA solution (Thermo Fisher Scientific).

We incubated HEK 293T cells in Dulbecco's modified Eagle medium (DMEM, Invitrogen) with 10% heat-inactivated fetal bovine serum (v/v, GIBCO) plus 1% penicillin/streptomycin. We transfected HEK 293T cells using Lipofectamine 3000 Reagent (Thermo Fisher Scientific) according to the manufacturer's instructions. We harvested cells 24–48 h post transfection for genotyping.

**Genome editing and karyotyping of iPSCs**. To manipulate the copy number of *APP* in the APP1.1 line, we pretreated iPSCs with 10 μM Rock inhibitor (Y-27632 dihydrochloride, Tocris) 2 h before nucleofection. We then dissociated iPSCs with Accutase (Thermo Fisher Scientific) into single-cell suspensions and filtered them through a 100-μm filter. We nucleofected $2.0 \times 10^6$ iPSCs with pSpCas9nVQR(BB)-2A-GFP constructs expressing sgRNA1 and sgRNA2 by using a Lonza Nucleofector 2b Device (Program B-016) and Human Stem Cell Nucleofector Kit 1 (Lonza).

After recovery in culture for 2 days, we isolated GFP-positive iPSCs by Accutase dissociation followed by FACS (FACS Aria IIa, BD Biosciences), plated them at 1000–2000 cells per 32-mm Geltrex-coated plate, and cultured them in E8 media containing 10 μM Rock inhibitor for 2 days. After 10–14 days, we manually selected individual iPSC clones. We cultured half of each clone in individual wells of 48-well plates and submitted the other half to genomic DNA extraction using QuickExtract DNA Extraction Solution (Epicentre) and screening for *EcoRI* disruption. We excluded the possibility of large deletions by generating ~5.2-kb PCR amplicons surrounding the *APP* exon 16 using primers (5′-GCACGTTGCC AGATCTTAATAACAG-3′ and 5′-CCTGCAGCAGTGAACATCAAGG-3′). We validated potentially edited clones by Sanger sequencing. Briefly, we generated

~1.2-kb PCR amplicons surrounding the target genomic region using primers (5′-AACAAAGCCCCAAAGTAGCAG-3′ and 5′- GCATGTGTTTGATCCTTCAA ATAAACC-3′), ligating them to a cloning vector, and subjecting them to Sanger sequencing. We quantitatively evaluated iPSC clones containing frameshifting mutations by next-generation deep sequencing of ~150-bp PCR amplicons using primers (5′-CAGGTTCTGGGTTGACAAATATCAAG-3′ and 5′-GCAAGACAA ACAGTAGTGGAAAGAGG-3′) and analyzed them using CRISPResso2[49]. The percentage of deep-sequencing reads for each iPSC clone indicated the number of *APP* copies remaining intact or removed.

We determined the karyotypes and genomic amplification and deletion regions of the parent and isogenic iPSCs by array-based CGH assay using Agilent Human Catalog CGH 8x60K Microarrays.

**Availability of iPSC lines**. The gene-edited iPSC lines with monoallelic, biallelic, or triallelic knockout of *APP* are available upon request from the Ip Laboratory.

**Differentiation of iPSCs into cortical neurons**. We generated cortical neurons from iPSCs as previously described[15,29] with minor modifications. On day −2, we plated iPSCs on six-well Geltrex-coated plates 1 day before infection. On day −1, we suspended lentivirus with rtTA and Ngn2-GFP-puro vectors in E8 media containing 10 μM Rock inhibitor and infected the iPSCs for 24 h. On day 0, to induce the expression of *NEUROG2* (the Ngn2 gene), we replaced the culture medium with N2 medium (DMEM/F12, 1× N2, and 1× NEAA; Thermo Fisher Scientific) containing 2 μg mL⁻¹ doxycycline (Sigma). On day 1, we replaced the culture medium with N2 medium containing 2 μg mL⁻¹ doxycycline and 1 μg mL⁻¹ puromycin (Thermo Fisher Scientific). On days 2–3, we replaced the medium with B27 medium (Neurobasal medium, 1× B27, and 1× GlutaMAX; Thermo Fisher Scientific) containing 2 μg mL⁻¹ doxycycline, 1 μg mL⁻¹ puromycin, and 0.2 μg mL⁻¹ laminin (Sigma). On day 4, we dissociated the induced cells with Accutase and plated them onto six-well PDL/laminin-coated plates at $1.0 \times 10^6$ cells per well or 24-well plates with coverslips at 50,000 cells per coverslip. For further experiments, until day 28, we removed half of the medium every 4 days and replaced it with B27 medium containing 0.2 μg mL⁻¹ laminin, 0.5 μg mL⁻¹ doxycycline, 10 ng mL⁻¹ BDNF, and 10 ng mL⁻¹ NT-3.

**Immunocytochemistry**. We fixed cells grown on coverslips or chamber slides with 4% paraformaldehyde (Electron Microscopy Services) for 15 min at room temperature. We then permeabilized the cells using 0.1% Triton X-100 in Dulbecco's phosphate-buffered saline (DPBS) for 15 min at room temperature, blocked them in 2% normal goat serum in DPBS for 1 h at room temperature, and incubated them with primary antibodies overnight at 4 °C. We then washed the cells three times in DPBS, incubated them with the appropriate secondary antibodies for 1 h at room temperature, washed them twice with DPBS, and incubated them with DAPI for 10 min. We acquired images with a Leica SP8 inverted confocal microscope.

**Protein extraction and western blot analysis**. We lysed cells with RIPA buffer containing protease inhibitor cocktail (Roche). We measured protein concentration by Bradford Assay (Bio-Rad). Briefly, we denatured samples at 95 °C for 5 min in sample buffer. We separated 20 μg protein per sample on 4–12% Bis-Tris or 10–20% Tricine gradient gels (Thermo Fisher Scientific) at 100 V for 1 h and subsequently transferred samples to nitrocellulose membranes (iBlot Transfer Stack, Thermo Fisher Scientific). We then used HRP-conjugated secondary antibodies and ECL substrate for detection.

**RNA extraction and RT-PCR**. We purified total RNA from iPSCs and iPSC-derived neurons by using the NucleoSpin RNA Kit (Macherey-Nagel) according to the manufacturer's instructions. We assessed RNA concentration and quality by using BioDrop μLite (BioDrop). We performed reverse-transcription by using a PrimeScript RT Reagent Kit (TaKaRa). We performed real-time qPCR with TaqMan gene expression assays and the Premix Ex Taq qPCR assay (TaKaRa). Finally, we used the 2exp (i.e., −ΔΔCt) method to determine the relative expression of each gene with HPRT as a reference.

**RNA sequencing analysis of iPSC-derived neurons**. We performed the transcript-level expression analysis as previously described[50]. Briefly, quality control was first accomplished using FastQC and RSeQC. The FASTQ files were aligned to the human GRCh38.91 reference genome using the HISAT2 2.0 alignment program. The aligned files were assembled into potential transcripts using the StringTie assembler program. Next, the potential transcripts were passed to the prepDE program to generate read counts. Minimal pre-filtering with row counts greater than ten reads was performed. We performed differential gene expression analysis using DESeq2 from Bioconductor version 3.10[51]. We used an FDR cutoff of 0.05 with a fold change ratio cutoff of 1.5 and a mean FPKM cutoff of 1 for differential gene expression analysis. We created the pipeline to run the analysis using the Snakemake workflow management system[52].

**Quantification of secreted Aβ**. We replaced the media 2 days before quantifying secreted Aβ. We quantified secreted Aβ by using MSD Human (6E10) Abeta3-Plex Kits (Meso Scale Discovery) according to the manufacturer's instructions. We normalized secreted Aβ levels to total protein levels determined by BCA assay (Thermo Fisher Scientific).

**Statistics and reproducibility**. All data are presented as mean ± SEM unless otherwise specified. We examined the significance of differences by unpaired Student's $t$-tests across three or four independent biological replicates using GraphPad prism version 6 (GraphPad Software). The level of statistical significance was set at $*p < 0.05$, $**p < 0.01$, or $***p < 0.001$.

**Reporting summary**. Further information on research design is available in the Nature Research Reporting Summary linked to this article.

## Data availability

The deep-sequencing data are deposited in the NCBI Sequence Read Archive (SRA; accession code: PRJNA681926). The RNA sequencing data are deposited in the NCBI Gene Expression Omnibus (GEO; accession code: GSE160224). The CGH data are available in Supplementary Data 1. The source data for the graphs and charts presented in the main figures are available in Supplementary Data 2. Full blots are included in Supplementary Fig. 6.

## Code availability

Custom code for genome-wide coverage analysis of various Cas9 nickase configurations in the GRCh38 human reference genome is available in Supplementary Software 1.

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

## Acknowledgements

We thank Dr. Tom H. Cheung, Dr. Wing-Yu Fu, Wayne Chi Wai Ng, Jingyi Zhang, Wei Wu, Dr. Carole Shum, Dr. Kwok-Wang Hung, Dr. Xiaopu Zhou, Dr. Nandita Mullapudi, Shun Fat Lau, Cara Kwong, Ka Chun Lok, Jingwen Lin, Hongruo Zhang, and Paul Chow for their excellent advice and technical assistance. We thank Dr. Lawrence Goldstein at the University of California, San Diego, for providing induced pluripotent stem cells. We are grateful to all members of the Ip Laboratory for their helpful discussions. This study was supported in part by the National Key R&D Program of China (2017YFE0190000 and 2018YFE0203600); the Hong Kong Research Grants Council Theme-based Research Scheme (T13-605/18-W); the Area of Excellence Scheme of the University Grants Committee (AoE/M-604/16); the Innovation and Technology Commission (ITCPD/17-9); the Lee Hysan Foundation (LHF17SC01); the Guangdong Provincial Key S&T Program (2018B030336001); the Shenzhen Knowledge Innovation Program (JCYJ20180507183642005, JCYJ20170413165053031, JCYJ20200109115631248, and JCYJ20170413173717055); and the Shenzhen–Hong Kong Institute of Brain Science-Shenzhen Fundamental Research Institutions (project number: 2019SHIBS0001).

## Author contributions

T.Y., A.K.Y.F., and N.Y.I. designed the research. T.Y., Y.D., H.W.S.T., H.X., Yuewen C., and Yu C. conducted the experiments. H.C. performed the bioinformatics analysis. T.Y., A.K.Y.F., and N.Y.I. analyzed the data. T.Y., A.K.Y.F., and N.Y.I. wrote the paper with input from all authors.

## Competing interests

The authors declare no competing interests.
