## [Peer Review File · Communications Biology]

Reviewers' comments:

Reviewer #1 (Remarks to the Author):

In this study, Ye et al. used a pair of Cas9nVQR, a variant Cas9n which recognizes NGA PAM instead of standard NGG PAM, to disrupt exon 16 of the human APP gene. They introduced these paired nickases into iPSCs derived from a patient of Alzheimer's disease (AD) carrying duplicated APP (three copies of the APP gene), and isolated cell clones undergoing monoallelic, biallelic, and triallelic APP disruption. They subsequently utilized one of the isolated iPSC clones that had undergone monoallelic APP disruption (i.e., carrying two functional APP alleles) as an APP-wildtype revertant, to explore the pathobiological consequences of APP duplication. By comparing isogenic sibling neurons differentiated from the APP-wildtype revertant and its parental AD iPSCs, respectively, together with non-demented healthy controls, the authors acquired a gene expression profile characteristic of APP duplication, and identified a set of genes whose expression were highly altered in APP-duplicated cells. One of those genes was PEG3, a gene implicated in p53-mediated cell death. The authors provided evidence that the PEG3 and its downstream BAX are involved in neuronal death resulting from APP duplication.

General Comments

1. Given the title of this manuscript, it appears that the authors would like to give emphasis on methodological advances achieved with genome manipulation conducted in this study. In regard to molecular tools used in this study for genome editing, a novel Cas9 variant, Cas9nVQR, was created by introducing VQR mutations (Kleinstiver et al. Nature 523, 481-5, 2015) together with a D10A mutation (Jinek et al. Science 337, 816-21, 2012) into the Cas9 protein. The use of paired Cas9 nickases for genome editing in human pluripotent stem cells has been reported in previous papers (e.g., Ran et al. Cell 154, 1380-9, 2013). Concurrent monoallelic and multiallelic editing of a gene using the CRISPR-Cas9 system is also not novel (e.g., Paquet et al. Nature 533, 125-9, 2016). The extent of advance may be an issue when this work is considered as a method paper.
2. The authors carried out Sanger-based genotyping of the APP gene in four candidate iPSC clones, and obtained three clones undergoing monoallelic, biallelic, and triallelic APP disruption, respectively (Figure 1d, e). Although they successfully did it in a single experiment, it actually depends on luck whether one can create all these clones at a time. Despite a general rule that method papers should describe new/improved methods that can be readily reproduced, it will probably be difficult to reproduce genome editing outcomes described in this study with a single experiment.
3. Genotyping of the manipulated APP gene in each iPSC clone was carried out solely based on Sanger sequencing of vectors in which PCR-amplified APP genomic locus had been cloned. Although the authors' experiment yielded surprisingly convincing genotyping results (e.g., 20 wildtype reads vs. 10 mutant reads for monoallelic disruption, and 10 wildtype reads vs. 20 mutant reads for biallelic disruption; Figure 1e), what is often anticipated in general is more ambiguous experimental results which modestly contribute to solid genotyping (e.g., 17 wildtype reads vs. 13 mutant reads). Accordingly, the authors may consider the examination of edited genomic site by additional assays such as NGS-based deep sequencing, digital PCR, and quantitative real-time PCR, in order to confirm the outcomes of Sanger-based genotyping.

Specific comments:

1. Figure 1d: Which of the four positive iPSC clones represent monoallelic, biallelic, and triallelic disruption of the APP gene?
2. Figure 3a: Why not include all of gene expression data for Iso, APP, and NDC into a single

hierarchical clustering analysis?

3. Figure 4e, f: The administration of BAX inhibitor followed by the staining of cleaved Caspase-3 may not be sufficient to provide convincing evidence for the involvement of PEG3 in cellular apoptosis elicited by APP duplication. It may be helpful to consider using additional methods, e.g., RNAi knockdown of BAX or PEG3, and cell staining with Annexin-V, 7-AAD, or Propidium Iodide, to provide supportive data.

4. Legend for Figure 4b: Data are not shown relative to those in APP neurons.

5. Supplementary Figure S3b: Resolution of image should be improved.

6. M&M 'Quantitative analysis': It is stated that unpaired Student's t-test and one-way ANOVA were used for statistical analyses in this study. The authors should specify the use of ANOVA within corresponding figure legends, just as they did for the use of t-test.

Reviewer #2 (Remarks to the Author):

Ye et al. reported a new IPS cell line carrying a duplication of APP with control cell lines derived from the original IPS using a new tool for genome editing based on a paired-nickases with a Cas9nVQR variant with a NGA PALM. They showed the impact of the genetic correction of APP dosage on amyloid and Tau-associated phenotypes. Then they described some cellular phenotypes, pathways and genes that are altered in the Dup(APP) cell line compared to control. The study is well presented and the new models are of interest if there are available to the scientific community.

Nevertheless, the authors should answer some major and minor concerns expressed below

Major:

1. The terms ISO cells should be clearly defined. When working with duplication on one chromosome and no duplication on the second homologous chromosome: (APP-APP)/APP, the removal of one functional allele can lead to 3 genetic configurations. Deleting one allele in a duplication is not the same as removing the single allele on the homologous chromosome and keeping the two duplicated alleles on the other. Indeed It would be nice to see if any difference is attached to the new iPS (APP-APP)/0 or (APP;KO)/APP or (KO;APP)/APP. And the situation is even more complex when doing two KO: (KO;KO)/APP, (APP;KO)/KO or (KO;APP)/KO. Somehow the figure 1a is oversimplified and misleading and should be adapted accordingly and the 2 new lines with 2 copies of APP, 1 copies of APP should be defined carefully not by the single number or copies and the noted APP-/- should be check for still carrying a duplication as only two alleles for APP have been detected. One could results from the deletion of the duplicated APP locus. Indeed the proposed experiments with theCas9VQR could lead also to the full deletion of the Duplicated allele leading to a cell line with different genotypes that could be Dup(DelAPP)/APP or Dup(delAPP)/KO (as reported different groups in other condition(BOROVIAK et al. 2016; BIRLING et al. 2017)). Thus it would be nice to show that the cell still carry a duplication without 3 functional copies of APP by CGH or qPCR outside of the APP exon 16 (upstream and downstream). The current Fig S3b can not be used to check for such changes.

2. Finally, in term of Iso line it will be important to detail the current allelic composition of APP in the original lines with 3 copies of APP (1,2 and 3). If 3 alleles are present, the term Iso cell line should be carefully defined. Here I can not understand if the Iso cell line is APP(1-2/KO), APP(1-KO/3 or (APP-KO-2/3? Or even with the duplication deleted APP(KO/3)?

3. Authors should carefully state where all the IPS cell lines are available for research

4. Can the authors propose some mechanism to explain how 3 copies of APP lead to an increase in P-TAU231/total? The origin of the IPS should be clearly defined (male or female carrier)

5. The authors described briefly the characterization of iPS derived neurons from Dup(APP), "Iso" and NDC iPS cells showing that they expressed neuron markers such as MAP2, NEUN, SYNAPTOPHYSIN (1 or 2?) and PDS95 but they do not show if iPS-derived neurons are functional with synaptic activity or blocked at any stage of differentiation. Can they provide further information?

6. The authors controlled a few genes that are differentially expressed, in particular MEG3 and PEG3. As both genes are imprinted long non-coding or protein coding RNA gene expressed respectively from the maternal or paternal genome. It would be important to clearly indicate the origin of both alleles which are differentially expressed and to be sure that the imprinting has not been affected by the genetic manipulation.

Minor

7. Correct « an NGA » page 5 lane 112

8. Please correct the statement page 3, lane 53: certain genes (e.g., ABCA7) confer a risk of late-onset AD termed "sporadic AD," which accounts for more than 90% of AD cases^{4, 5}.

9. Fig S3b should be bigger to allow allowing the reader to see all the details.Ips

Reference

Birling, M. C., L. Schaeffer, P. André, L. Lindner, D. Maréchal et al., 2017 Efficient and rapid generation of large genomic variants in rats and mice using CRISMERE. *Sci Rep* 7: 43331.

Boroviak, K., B. Doe, R. Banerjee, F. Yang and A. Bradley, 2016 Chromosome engineering in zygotes with CRISPR/Cas9. *Genesis* 54: 78-85.

Reviewer #3 (Remarks to the Author):

Brief Summary:

In this manuscript, the authors employed paired Cas9 nickases to create a double-strand breakage in APP 16 exon and to isolate Iso cells with a deletion of an additional APP gene in a patient-derived iPSC line. They were able to isolate cells with monoallelic, biallelic, or 38 triallelic knockout of APP. The gene-corrected Iso cells were differentiated into cortical neurons from which the pathological conditions were recovered to a normal state, when compared to NDC. They finally claimed that an APP-PEG3-Bax axis that promotes neuronal death in a human iPSC model of familial AD carrying APP duplication.

Overall impression of the work

Overall work in this study can be divided into two sections: One is to use CRISPR technology to derive Iso cells from APP cells and the other to investigate molecular events surrounding AD pathology induced by APP duplication. Though the authors used the term "One-step selection" and "Scarless" in the manuscript, this study did not provide any technical improvement appropriate for such use. Additionally, the later part on AD pathology was not supported by sufficient evidence on signal transduction pathway and validations by another round of gene correction as mentioned in the specific comments.

Specific Comments

Comment 1:

None of the methods employed in this manuscript are new. Ever since the double-nicking DSB system was first reported by Guilinger et al., (Biotechnology 2014), lots of studies have relied on this system. Moreover, using Cas9-NG is also proven in many other works (eg, Zhang et al., Nat Commun), just combining two methods is never fresh. Thus, I can't find any incorporation of new technological improvement in this study.

Comment 2:

The approach adopted in this work is just inducing indel mutation rather than site-specific gene correction. A broader spectrum of APP exon 16 is likely to contain another NGG PAM motif. Just working with a wild type Cas9 is enough for this aim. Do you have any reason for using Cas9-NG variant? The presence of EcoRI site between the targets is the reason? Rather than using a PAM variant of Cas9 and double-nicking system, more precise versions of Cas9 (eg. eSp-Cas9, HF-Cas9, etc.) would be preferred.

Comment 3:

In Fig. 1b, the efficiency was tested in HEK293T cells, not iPSCs. The efficiency between the two cells is quite different. The actual efficiency should be provided through more quantitative measurements, like NGS-based deep sequencing. A simple validation of gRNAs is meaningless in Fig. 1b.

Comment 4:

The first result section begins with the expression "Scarless manipulation of APP copy number..". However, I don't understand what the 'Scarless' implies. Do you mean by "no off-target"? Then, the authors did not provide any evidence on off-target level through either a genome-wide or targeted method.

Comment 5:

The title of this manuscript also begins with "One-step manipulation of gene..". It is usual that most indel works are done in a one-step manner by using efficient gRNAs. Unless you selected the Iso iPSCs in a one-step-selection manner, the expression "One-step" would not be appropriate for this work.

Comment 6:

For real cohorts of AD patients, it is likely that each copy of APP gene shows different expression level, which may mean that a locus-specific manipulation of gene dosage is required. If you can develop a method that meets such requirements, it would add up attractions to your study.

Comment 7:

Because the authors are likely to transfect multiple cells at the transfection stage and also to prepare genomic DNA from multiple cells, the authors did not evaluate the paired nickase system at a single-cell level in this manuscript, as the authors claimed in line 95.

Comment 8:

In relation to comment 6, I would like to stress the importance of selecting a specific copy of APP gene for deletion. In Fig. 3a, the authors used three batches of APP neurons and the three batches are preferred to contain three different lines, each of which shows deletion at each different APP-locus.

Comment 9:

The authors claimed that an APP-PEG3-Bax axis that promotes neuronal death in a human iPSC model of familial AD carrying APP duplication. To support this, they relied on the Caspase 3 activation but failed to provide additional evidence by investigating downstream players and

events associated the PEG3-Bax axis. To support their claim, a more rigorous study needs to be complemented.

Comment 10:

Finally, if a deleted copy of APP gene is responsible for the restoration of pathogenic conditions around AD, then it is highly required to prove that the correction of deleted APP gene to a wild-type sequence in Iso cells again induces pathogenic conditions. This step will support the authors' claim in a more solid manner. The prime editing tool (Anzalone et al., Nature 2019) would be recommended for such correction.

Point-by-point Responses to Reviewers
COMMSBIO-20-0728-T

Reviewer #1 (Remarks to the Author):

In this study, Ye et al. used a pair of Cas9nVQR, a variant Cas9n which recognizes NGA PAM instead of standard NGG PAM, to disrupt exon 16 of the human APP gene. They introduced these paired nickases into iPSCs derived from a patient of Alzheimer's disease (AD) carrying duplicated APP (three copies of the APP gene), and isolated cell clones undergoing monoallelic, biallelic, and triallelic APP disruption. They subsequently utilized one of the isolated iPSC clones that had undergone monoallelic APP disruption (i.e., carrying two functional APP alleles) as an APP-wildtype revertant, to explore the pathobiological consequences of APP duplication. By comparing isogenic sibling neurons differentiated from the APP-wildtype revertant and its parental AD iPSCs, respectively, together with non-demented healthy controls, the authors acquired a gene expression profile characteristic of APP duplication, and identified a set of genes whose expression were highly altered in APP-duplicated cells. One of those genes was PEG3, a gene implicated in p53-mediated cell death. The authors provided evidence that the PEG3 and its downstream BAX are involved in neuronal death resulting from APP duplication.

General Comments

1. Given the title of this manuscript, it appears that the authors would like to give emphasis on methodological advances achieved with genome manipulation conducted in this study. In regard to molecular tools used in this study for genome editing, a novel Cas9 variant, Cas9nVQR, was created by introducing VQR mutations (Kleinstiver et al. *Nature* 523, 481-5, 2015) together with a D10A mutation (Jinek et al. *Science* 337, 816–21, 2012) into the Cas9 protein. The use of paired Cas9 nickases for genome editing in human pluripotent stem cells has been reported in previous papers (e.g., Ran et al. *Cell* 154, 1380-9, 2013). Concurrent monoallelic and multiallelic editing of a gene using the CRISPR-Cas9 system is also not novel (e.g., Paquet et al. *Nature* 533, 125-9, 2016). The extent of advance may be an issue when this work is considered as a method paper.

We thank the reviewer for the comment. Compared to previous studies, the method described in our manuscript represents 3 methodological advancements:

1) Using paired Cas9 nickases for genome editing significantly enhances specificity albeit at the cost of restricted genome targetability (e.g., Ran et al., *Cell* 154, 1380–1389, 2013); this limitation must be overcome in order to broaden applications in disease modeling. To our knowledge, our work is the first to expand the genome-targeting range of nickase-mediated double-nicking in human induced pluripotent stem cells (iPSCs).

2) Although VQR mutations (Kleinstiver et al., *Nature* 523, 481–485, 2015) and D10A mutation (Jinek et al., *Science* 337, 816–821, 2012) have been generated individually, our work is the first to simultaneously introduce them to create a Cas9nVQR variant and demonstrate its effectiveness in paired nickase-mediated genome editing, specifically generating a large genomic deletion of the 722-kb duplicate region harboring the *APP* locus (**new Supplementary Table S3**).

3) The currently used method for concurrent mono- and multiallelic gene editing (e.g., Paquet et al., *Nature* 533, 125–129, 2016) requires 2 sequential rounds of homology-directed repair-based knock-in, which is less efficient and takes at least twice as long as our method. In contrast, our method can efficiently, rapidly, and simultaneously generate iPSC lines with mono-, bi-, or triallelic knockout of a gene with a single editing step and substantially less effort. This is particularly important for rapidly generating and modeling disease risk genes in human iPSC models. Furthermore, similar approaches applying the same principle can further expand genome targetability using other PAM variants such as xCas9 (Hu et al., *Nature* 556, 57-63, 2018) and SpRY (Walton et al., *Science* 368, 290-296, 2020).

Accordingly, we have incorporated these points in the *Discussion*.

2. The authors carried out Sanger-based genotyping of the *APP* gene in four candidate iPSC clones, and obtained three clones undergoing monoallelic, biallelic, and triallelic *APP* disruption, respectively (Figure 1d, e). Although they successfully did it in a single experiment, it actually depends on luck whether one can create all these clones at a time. Despite a general rule that method papers should describe new/improved methods that can be readily reproduced, it will probably be difficult to reproduce genome editing outcomes described in this study with a single experiment.

We thank the reviewer for the constructive comment. We have conducted an additional experiment to demonstrate that the genome editing outcomes can be reproduced in a single experiment. We repeated the editing of the *APP* exon 16 site and generated 4 new clones. As shown below, clones 2 and 3 underwent mono- and biallelic *APP* disruption, respectively, while clones 12 and 14 both underwent triallelic *APP* disruption. We did not further characterize clones 8 and 11, which had in-frame truncation of *APP*. These new replicates demonstrate the reproducibility of the method described in our manuscript.

3. Genotyping of the manipulated *APP* gene in each iPSC clone was carried out solely based on Sanger sequencing of vectors in which PCR-amplified *APP* genomic locus had been cloned. Although the authors' experiment yielded surprisingly convincing genotyping results (e.g., 20 wildtype reads vs. 10 mutant reads for monoallelic disruption, and 10 wildtype reads vs. 20 mutant reads for biallelic disruption; Figure 1e), what is often anticipated in general is more ambiguous experimental results which modestly contribute to solid genotyping (e.g., 17 wildtype reads vs. 13 mutant reads). Accordingly, the authors may consider the examination of edited genomic site by additional assays such as NGS-based deep sequencing, digital PCR, and quantitative real-time PCR, in order to confirm the outcomes of Sanger-based genotyping.

We thank the reviewer for the constructive comment. Accordingly, we have performed additional next-generation deep sequencing to genotype the manipulated *APP* in each iPSC clone and replaced the Sanger-based genotyping results (Figure 1e and Supplementary **Figure S2**). The next-generation deep sequencing results show 71% wild-type (WT) reads versus 29% mutant reads for monoallelic disruption, 52% WT reads versus 48% mutant reads for biallelic disruption, and 49% 16 bp deletion reads versus 51% 11 bp deletion reads for triallelic disruption (**Figure 1e**). In addition, we conducted an array-based comparative genomic hybridization analysis. The array-based comparative genomic hybridization data for chromosome 21 show that the edited iPSC lines with monoallelic disruption still carry a 722-kb duplication like the parent *APP* duplication iPSC line, whereas the iPSC lines with bi- or triallelic disruption do not (**new Supplementary Table S3**). These results collectively suggest that paired Cas9nVQR deletes the entire 722-kb duplicate region harboring the *APP* locus in iPSC lines with bi- or triallelic disruption, which potentially represents one of the largest reported double-nicking-mediated genomic deletions.

Specific comments:

1. Figure 1d: Which of the four positive iPSC clones represent monoallelic, biallelic, and triallelic disruption of the *APP* gene?

We have labeled the 3 positive iPSC clones—clones 9, 1, and 11—which represent mono-, bi-, and triallelic disruption of *APP*, respectively, in **Figure 1e**. As we identified an 18-bp

in-frame truncation in the fourth iPSC clone (clone 2) using Sanger sequencing, we did not further characterize this clone.

2. Figure 3a: Why not include all of gene expression data for Iso, APP, and NDC into a single hierarchical clustering analysis?

We thank the reviewer for the constructive comment. Accordingly, we have redone the analysis and included all gene expression data for Iso, APP, and NDC neurons in a single hierarchical clustering analysis in **Figure 3a**.

3. Figure 4e, f: The administration of BAX inhibitor followed by the staining of cleaved Caspase-3 may not be sufficient to provide convincing evidence for the involvement of PEG3 in cellular apoptosis elicited by APP duplication. It may be helpful to consider using additional methods, e.g., RNAi knockdown of BAX or PEG3, and cell staining with Annexin-V, 7-AAD, or Propidium Iodide, to provide supportive data.

We thank the reviewer for the constructive comment. As suggested, we have performed an additional experiment involving cell staining with propidium iodide and Hoechst for apoptosis and included the new data (**new Supplementary Figure S4**). The RNAi knockdown of BAX or PEG3 in iPSC-derived neurons at 21 days in vitro is technical challenging. Nonetheless, we performed additional experiments to show that blockade of cytochrome C release in APP iPSC-derived neurons by the treatment with Bax channel blocker significantly reduced cleaved Caspase-3 and hence neuronal apoptosis (**Figure 4e, f**).

4. Legend for Figure 4b: Data are not shown relative to those in APP neurons.

We thank the review for pointing this out. Accordingly, we have revised “APP” to “NDC” neurons in the legend for **Figure 4b**.

5. Supplementary Figure S3b: Resolution of image should be improved.

We thank the review for pointing this out. Accordingly, we have enlarged the image and improved the resolution of **Figure S3b**.

6. M&M ‘Quantitative analysis’: It is stated that unpaired Student’s t-test and one-way ANOVA were used for statistical analyses in this study. The authors should specify the use of ANOVA within corresponding figure legends, just as they did for the use of t-test.

We thank the reviewer for pointing this out. We used unpaired Student’s *t*-tests for all statistical analyses in our study. We revised the *Quantitative analysis* section in the *Methods* accordingly.

Reviewer #2 (Remarks to the Author):

Ye et al. reported a new IPS cell line carrying a duplication of APP with control cell lines derived from the original IPS using a new tool for genome editing based on a paired-nickases with a Cas9nVQR variant with a NGA PALM. They showed the impact of the genetic correction of APP dosage on amyloid and Tau-associated phenotypes. Then they described some cellular phenotypes, pathways and genes that are altered in the Dup(APP) cell line compared to control. The study is well presented and the new models are of interest if there are available to the scientific community.

Nevertheless, the authors should answer some major and minor concerns expressed below

Major:

1. The terms ISO cells should be clearly defined. When working with duplication on one chromosome and no duplication on the second homologous chromosome: (APP-APP)/APP, the removal of one functional allele can lead to 3 genetic configurations. Deleting one allele in a duplication is not the same as removing the single allele on the homologous chromosome and keeping the two duplicated alleles on the other. Indeed It would be nice to see if any difference is attached to the new iPS (APP-APP)/0 or (APP;KO/APP) or (KO;APP)/APP. And the situation is even more complex when doing two KO: (KO;KO/APP), (APP;KO/KO) or (KO;APP)/KO. Somehow the figure 1a is oversimplified and misleading and should be adapted accordingly and the 2 new lines with 2 copies of APP, 1 copies of APP should be defined carefully not by the single number or copies and the noted APP-/-/- should be check for still carrying a duplication as only two alleles for APP have been detected. One could results from the deletion of the duplicated APP locus. Indeed the proposed experiments with the Cas9VQR could lead also to the full deletion of the Duplicated allele leading to a cell line with different genotypes that could be Dup(DelAPP)/APP or Dup(delAPP)/KO (as reported different groups in other condition (BOROVIK et al. 2016; BIRLING et al. 2017). Thus it would be nice to show that the cell still carry a duplication without 3 functional copies of APP by CGH or qPCR outside of the APP exon 16 (upstream and downstream). The current Fig S3b can not be used to check for such changes.

We thank the reviewer for the constructive feedback. Accordingly, we have revised the *Introduction* to clearly define “Iso cells” as an edited iPSC line that has undergone monoallelic *APP* disruption (i.e., carrying 2 functional *APP* alleles) as an *APP* WT revertant. As suggested, we have modified the model in **Figure 1a** to account for *APP* alleles and chromosome duplication. We have also provided array-based comparative genomic hybridization data for chromosome 21 showing that the edited iPSC lines with monoallelic disruption still carry a 722-kb duplication like the parent *APP* duplication iPSC line, whereas the iPSC lines with bi- or triallelic disruption do not (**new Supplementary Table 3**). These results collectively suggest that paired Cas9nVQR deletes the entire 722-kb duplicate region harboring the *APP* locus in iPSC lines with bi- or triallelic disruption, which potentially represents one of the largest reported double-nicking-mediated genomic deletions.

2. Finally, in term of Iso line it will be important to detail the current allelic composition of APP in the original lines with 3 copies of APP (1,2 and 3). If 3 alleles are present, the term Iso cell line should be carefully defined. Here I can not understand if the Iso cell line is APP(1-2/KO), APP(1-K0/3 or (APP-KO-2/3? Or even with the duplication deleted APP(KO/3)?

We thank the reviewer for raising this issue. The original *APP* duplication iPSC line harbors a relatively large 722-kb duplicate genomic segment containing the *APP* gene that is retained in Iso cells (**new Supplementary Table 3**) (Rovelet-Lecrux et al., *J Neurol Neurosurg Psychiatry* 78, 1158-1159, 2007). Given that 3 copies of *APP* contain the same genomic sequence with respect to exon 16 for genome editing, it would be technically impossible to distinguish the duplicate *APP* copy from the normal copies.

3. Authors should carefully state where all the IPS cell lines are available for research

We thank the reviewer for pointing this out. Accordingly, we have included a section titled *Availability of iPSC lines* in the *Methods* stating that the gene-edited iPSC lines are available from the Ip Laboratory upon request.

4. Can the authors propose some mechanism to explain how 3 copies of APP lead to an increase in P-TAU231/total? The origin of the IPS should be clearly defined (male or female carrier)

We thank the reviewer for the insightful question. Indeed, in their original *Nature* paper, Israel et al. report that having 3 copies of *APP* leads to an increase in the p-Tau231/total Tau ratio compared to NDC neurons (Israel et al., *Nature* 482, 216-220, 2012). Meanwhile, we showed that the increase in the p-Tau231/total Tau ratio can be ameliorated by removing 1 copy of *APP*. As the reviewer suggested, we have added discussion about the possible mechanism by which increased *APP* expression leads to an increase in the p-Tau231/total Tau ratio via cholesterol metabolism, cholesteryl esters, and proteasome degradation with reference to a follow-up paper by Goldstein's group (van der Kant et al., *Cell Stem Cell* 24, 363-375.e369, 2019).

In addition, we have clearly stated the sex of the origin of the iPSC lines (i.e., “from male donors”) in the *Cell lines and cell culture* section in the *Methods*.

5. The authors described briefly the characterization of iPS derived neurons form Dup(*APP*), “Iso” and NDC iPS cells showing that they expressed neuron markers such as MAP2, NEUN, SYNAPTOPHYSIN (1 or 2?) and PDS95 but they do not show if iPS-derived neurons are functional with synaptic activity or blocked at any stage of differentiation. Can they provide further information?

We thank the reviewer for raising this issue. The primary aims of our study were to establish an efficient method for manipulating gene dosages and perform a proof-of-concept study in

human iPSCs. As such, examining synaptic activity or relevant phenotypes requires co-culture with astrocytes or astrocyte-conditioned medium (Wen et al., *Nature* 515, 414-418, 2014), which is outside the scope of the current study.

6. The authors controlled a few genes that are differentially expressed, in particular MEG3 and PEG3. As both genes are imprinted long non-coding or protein coding RNA gene expressed respectively from the maternal or paternal genome. It would be important to clearly indicate the origin of both alleles which are differentially expressed and to be sure that the imprinting has not been affected by the genetic manipulation.

We thank the reviewer for the comment. As we detected no allele-specific single nucleotide polymorphisms (SNPs) in *PEG3* transcripts by RNA sequencing, it would be technically challenging to determine the origin of the *PEG3* expression induced in APP neurons and also outside the scope of the current study. As its name suggests, *PEG3* (*paternally expressed 3*) is likely expressed from the paternal genome in *APP* duplication neurons.

Minor

7. Correct « an NGA » page 5 lane 112

We thank the reviewer for pointing this out and have corrected this in the manuscript accordingly.

8. Please correct the statement page 3, lane 53: certain genes (e.g., ABCA7) confer a risk of late-onset AD termed “sporadic AD,” which accounts for more than 90% of AD cases^{4, 5}.

We thank the reviewer for pointing this out and have corrected the statement in question in the manuscript accordingly.

9. Fig S3b should be bigger to allow allowing the reader to see all the details.Ips

We thank the reviewer for the helpful suggestion. Accordingly, we have enlarged the image and improved the resolution of **Figure S3b**.

Reference

Birling, M. C., L. Schaeffer, P. André, L. Lindner, D. Maréchal et al., 2017 Efficient and rapid generation of large genomic variants in rats and mice using CRISMERE. *Sci Rep* 7: 43331.

Boroviak, K., B. Doe, R. Banerjee, F. Yang and A. Bradley, 2016 Chromosome engineering in zygotes with CRISPR/Cas9. *Genesis* 54: 78-85.

Reviewer #3 (Remarks to the Author):

Brief Summary:

In this manuscript, the authors employed paired Cas9 nickases to create a double-strand breakage in APP 16 exon and to isolate Iso cells with a deletion of an additional APP gene in a patient-derived iPSC line. They were able to isolate cells with monoallelic, biallelic, or 38 triallelic knockout of APP. The gene-corrected Iso cells were differentiated into cortical neurons from which the pathological conditions were recovered to a normal state, when compared to NDC. They finally claimed that an APP–PEG3–Bax axis that promotes neuronal death in a human iPSC model of familial AD carrying APP duplication.

Overall impression of the work

Overall work in this study can be divided into two sections: One is to use CRISPR technology to derive Iso cells from APP cells and the other to investigate molecular events surrounding AD pathology induced by APP duplication. Though the authors used the term “One-step selection” and “Scarless” in the manuscript, this study did not provide any technical improvement appropriate for such use. Additionally, the later part on AD pathology was not supported by sufficient evidence on signal transduction pathway and validations by another round of gene correction as mentioned in the specific comments.

Specific Comments

Comment 1:

None of the methods employed in this manuscript are new. Ever since the double-nicking DSB system was first reported by Guilinger et al., (Biotechnology 2014), lots of studies have relied on this system. Moreover, using Cas9-NG is also proven in many other works (eg, Zhang et al., Nat Commun), just combing two methods is never fresh. Thus, I can't find any incorporation of new technological improvement in this study.

We thank the reviewer for the comment. Compared to previous studies, the method described in our manuscript represents 3 methodological advancements:

1) Using paired Cas9 nickases for genome editing significantly enhances specificity albeit at the cost of restricted genome targetability (e.g., Ran et al., *Cell* 154, 1380–1389, 2013); this limitation must be overcome in order to broaden applications in disease modeling. To our knowledge, our work is the first to expand the genome-targeting range of nickase-mediated double-nicking in iPSCs.

2) Although VQR mutations (Kleinstiver et al., *Nature* 523, 481–485, 2015) and D10A mutation (Jinek et al., *Science* 337, 816–821, 2012) have been generated individually, our work is the first to simultaneously introduce them to create a Cas9nVQR variant and demonstrate its effectiveness in paired nickase-mediated genome editing, specifically generating a large genomic deletion of the 722-kb duplicate region harboring the *APP* locus (**new Supplementary Table S3**).

3) The currently used method for concurrent mono- and multiallelic gene editing (e.g., Paquet

et al., *Nature* 533, 125–129, 2016) requires 2 sequential rounds of homology-directed repair-based knock-in, which is less efficient and takes at least twice as long as our method. In contrast, our method can efficiently, rapidly, and simultaneously generate iPSC lines with mono-, bi-, or triallelic knockout of a gene with a single editing step and substantially less effort. This is particularly important for rapidly generating and modeling disease risk genes in human iPSC models. Furthermore, similar approaches applying the same principle can further expand genome targetability using other PAM variants such as xCas9 (Hu et al., *Nature* 556, 57-63, 2018) and SpRY (Walton et al., *Science* 368, 290-296, 2020).

Accordingly, we have incorporated these points into the *Discussion*.

Comment 2:

The approach adopted in this work is just inducing indel mutation rather than site-specific gene correction. A broader spectrum of APP exon 16 is likely to contain another NGG PAM motif. Just working with a wild-type Cas9 is enough for this aim. Do you have any reason for using Cas9-NG variant? The presence of EcoRI site between the targets is the reason? Rather than using a PAM variant of Cas9 and double-nicking system, more precise versions of Cas9 (eg. eSp-Cas9, HF-Cas9, etc.) would be preferred.

We thank the reviewer for raising this issue. Indeed, a broader spectrum of *APP* exon 16 contains pairs of NGG PAM motifs that have a large offset (>20 bp) between the 2 guide RNAs spanning intron–exon junctions. Indels generated from WT Cas9 are likely to disrupt the splicing site between intron 15 and exon 16, resulting in exon 16 skipping and a truncated protein rather than knockout of the allele. By using a novel Cas9nVQR mutant, we located 2 NGA PAM motifs within *APP* exon 16 and generated indels within this exon in order to knock out the protein expression from the edited allele. As 2 NGG PAM motifs are not usually present in protein-coding exons, introducing an NGA PAM variant (i.e., Cas9nVQR) will greatly expand the targeting scope of the double-nicking system within the target exon for gene knockout while avoiding unwanted disruption of splicing events. Like WT Cas9, the more-precise versions of Cas9 (e.g., eSp-Cas9, HF-Cas9, etc.) would mostly result in biallelic or complete gene knockout owing to their high efficiency but would not overcome the challenge of simultaneously isolating heterozygous clones.

Comment 3:

In Fig. 1b, the efficiency was tested in HEK293T cells, not iPSCs. The efficiency between the two cells is quite different. The actual efficiency should be provided through more quantitative measurements, like NGS-based deep sequencing. A simple validation of gRNAs is meaningless in Fig. 1b.

We thank the reviewer for the constructive comment. Accordingly, we found that paired Cas9nVQR with sgRNA1 and sgRNA2 generated an editing efficiency of $40 \pm 3.5\%$ in iPSCs after GFP sorting through NGS-based deep sequencing. As suggested, we have included this information in the manuscript.

Comment 4:

The first result section begins with the expression “Scarless manipulation of APP copy number..”. However, I don’t understand what the ‘Scarless’ implies. Do you mean by “no off-target”? Then, the authors did not provide any evidence on off-target level through either a genome-wide or targeted method.

We thank the reviewer for raising this issue. “Scarless” implies footprint-free editing at the target site. Accordingly, we have clarified this by changing the subtitle of the first result and Figure 1 to “footprint-free” to differentiate our method from most existing methods that incorporate a selection marker for the enrichment of edited clones and leave an editing footprint at the target site (as summarized in **Supplementary Table S2**). Specifically, removing the footprint would require a second step of genetic manipulation.

Comment 5:

The title of this manuscript also begins with “One-step manipulation of gene..”. It is usual that most indel works are done in an one-step manner by using efficient gRNAs. Unless you selected the Iso iPSCs in a one-step-selection manner, the expression “One-step” would not be appropriate for this work.

We thank the reviewer for raising this issue. Because of the high efficiency of Cas9, efficient gRNAs mostly generate indels in all copies of the target gene (Canver et al., *The Journal of biological chemistry* 289, 21312-21324, 2014; Dow et al., *Nature biotechnology* 33, 390-394, 2015), which raises the challenge of isolating iPSCs with a single copy removed in “one step.” In order to rapidly and efficiently isolate iPSC lines with different dosages, we exploited the more-stringent criteria of the double-nicking system despite its moderately attenuated editing efficiency. As the reviewer suggested, we have revised the title of our report to, “*Efficient manipulation of gene dosage using CRISPR/Cas9 nickases in human iPSCs for disease modeling.*”

Comment 6:

For real cohorts of AD patients, it is likely that each copy of APP gene shows different expression level, which may mean that a locus-specific manipulation of gene dosage is required. If you can develop a method that meets such requirements, it would add up attractions to your study.

We thank the reviewer for the constructive comment. As CRISPR/Cas9 recognizes its target in a sequence-specific manner, locus- or allele-specific editing must rely on genetic variation or mutation in different copies of the target gene. As we detected no genetic variation in the target *APP* sequence, it would be impossible to manipulate *APP* exon 16 in a locus-specific manner. However, if there is genetic variation or mutation in the target gene, it would be feasible to achieve locus-specific manipulation of the allele containing a WT or mutant site by using allele-specific guide RNA.

Comment 7:

Because the authors are likely to transfect multiple cells at the transfection stage and also to prepare genomic DNA from multiple cells, the authors did not evaluate the paired nickase system at a single-cell level in this manuscript, as the authors claimed in line 95.

We thank the reviewer for raising this issue. As we evaluated the paired nickase system in single cell-derived clones, we have revised the term in question to “single cell-derived clone level” in the *Introduction*.

Comment 8:

In relation to comment 6, I would like to stress the importance of selecting a specific copy of APP gene for deletion. In Fig. 3a, the authors used three batches of APP neurons and the three batches are preferred to contain three different lines, each of which shows deletion at each different APP-locus.

We thank the reviewer for the comment. As we detected no genetic variation in the target sequence, it would be impossible to manipulate *APP* exon 16 in a locus-specific manner, making it difficult to specifically generate 3 lines each with deletion at a different *APP* locus.

Comment 9:

The authors claimed that an APP–PEG3–Bax axis that promotes neuronal death in a human iPSC model of familial AD carrying APP duplication. To support this, they relied on the Caspase 3 activation but failed to provide additional evidence by investigating downstream players and events associated the PEG3–Bax axis. To support their claim, a more rigorous study needs to be complemented.

We thank the reviewer for the constructive comment. As Bax translocation induces the release of cytochrome C during apoptosis (Pawlowski and Kraft, *Proc Natl Acad Sci U S A* 97, 529-531, 2000), we performed additional experiments, which showed that blockade of cytochrome C release in APP iPSC-derived neurons by treatment with Bax channel blocker significantly reduced the percentage of cleaved caspase 3⁺ neurons (by 10%) compared to the vehicle-treated condition (**Figure 4e, f**). This additional evidence suggests that cytochrome C acts downstream of the APP–PEG3–Bax axis in the mediation of neuronal apoptosis.

Comment 10:

Finally, if a deleted copy of APP gene is responsible for the restoration of pathogenic conditions around AD, then it is highly required to prove that the correction of deleted APP gene to a wild-type sequence in Iso cells again induces pathogenic conditions. This step will support the authors’ claim in a more solid manner. The prime editing tool (Anzalone et al., *Nature* 2019) would be recommended for such correction.

The prime editing tool has been rigorously tested for targeted point mutations and small insertions in HEK 293T cells but not iPSCs (Anzalone et al., *Nature* 576, 149-157, 2019). Human iPSCs are difficult to transfect and exhibit much lower post-editing transfection efficiency and survival than HEK 293T cells. In addition, the 23-bp deletion in the Iso iPSCs

is relatively long. Therefore, it would be technically challenging to correct the deleted *APP* gene to a WT sequence in Iso iPSCs by using the prime editing tool. Although establishing such a method to precisely and efficiently knock-in large insertions in iPSCs would be invaluable, it is beyond the scope of the current study.

References:

- Anzalone, A.V., Randolph, P.B., Davis, J.R., Sousa, A.A., Koblan, L.W., Levy, J.M., Chen, P.J., Wilson, C., Newby, G.A., Raguram, A., and Liu, D.R. (2019). Search-and-replace genome editing without double-strand breaks or donor DNA. *Nature* 576, 149-157.
- Canver, M.C., Bauer, D.E., Dass, A., Yien, Y.Y., Chung, J., Masuda, T., Maeda, T., Paw, B.H., and Orkin, S.H. (2014). Characterization of genomic deletion efficiency mediated by clustered regularly interspaced short palindromic repeats (CRISPR)/Cas9 nuclease system in mammalian cells. *The Journal of biological chemistry* 289, 21312-21324.
- Dow, L.E., Fisher, J., O'Rourke, K.P., Muley, A., Kasthuber, E.R., Livshits, G., Tschaharganeh, D.F., Socci, N.D., and Lowe, S.W. (2015). Inducible in vivo genome editing with CRISPR-Cas9. *Nature biotechnology* 33, 390-394.
- Hu, J.H., Miller, S.M., Geurts, M.H., Tang, W., Chen, L., Sun, N., Zeina, C.M., Gao, X., Rees, H.A., Lin, Z., and Liu, D.R. (2018). Evolved Cas9 variants with broad PAM compatibility and high DNA specificity. *Nature* 556, 57-63.
- Israel, M.A., Yuan, S.H., Bardy, C., Reyna, S.M., Mu, Y., Herrera, C., Hefferan, M.P., Van Gorp, S., Nazor, K.L., Boscolo, F.S., *et al.* (2012). Probing sporadic and familial Alzheimer's disease using induced pluripotent stem cells. *Nature* 482, 216-220.
- Pawlowski, J., and Kraft, A.S. (2000). Bax-induced apoptotic cell death. *Proc Natl Acad Sci U S A* 97, 529-531.
- Rovelet-Lecrux, A., Frebourg, T., Tuominen, H., Majamaa, K., Campion, D., and Remes, A.M. (2007). APP locus duplication in a Finnish family with dementia and intracerebral haemorrhage. *J Neurol Neurosurg Psychiatry* 78, 1158-1159.
- van der Kant, R., Langness, V.F., Herrera, C.M., Williams, D.A., Fong, L.K., Leestemaker, Y., Steenvoorden, E., Rynearson, K.D., Brouwers, J.F., Helms, J.B., *et al.* (2019). Cholesterol Metabolism Is a Druggable Axis that Independently Regulates Tau and Amyloid- β in iPSC-Derived Alzheimer's Disease Neurons. *Cell Stem Cell* 24, 363-375.e369.
- Walton, R.T., Christie, K.A., Whittaker, M.N., and Kleinstiver, B.P. (2020). Unconstrained genome targeting with near-PAMless engineered CRISPR-Cas9 variants. *Science* 368, 290-296.
- Wen, Z., Nguyen, H.N., Guo, Z., Lalli, M.A., Wang, X., Su, Y., Kim, N.S., Yoon, K.J., Shin, J., Zhang, C., *et al.* (2014). Synaptic dysregulation in a human iPSC cell model of mental disorders. *Nature* 515, 414-418.

Reviewers' comments:

Reviewer #1 (Remarks to the Author):

The authors extensively revised the manuscript supplementing several new experimental data. This reviewer finds significant improvement in the manuscript, nonetheless, has critiques and recommendations as listed below.

Re: General Comments in the first-round review

1. Extent of methodological advance:

Although this reviewer does not disagree with the expanded genome targetability achieved by the combined use of Cas9 (D10A) and VQR mutation compared to the use of Cas9 (D10A) nickase pairs, in this reviewer's opinion, genome editing in this study was achieved simply via the joint use of a few pre-existing technologies and there is no notable synergistic effect made by their joint use. In addition, there is no surprise in the creation of 722-kb deletion when two pairs of complementary nicks are introduced at the same location on the tandem duplication of (probably) 722-kb DNA region. Ran et al. have already created genomic deletions using two Cas9 nickase pairs, although the deletions were smaller (~ 6 kb; Ran et al. *Cell*, 154: 1380–9, 2013). Furthermore, it is likely a common knowledge that both mono- and multiallelic knockouts can be generated when an inefficient Cas9 nuclease against an autosomal gene is introduced into cells. This study appears to simply reproduce the same result using a nickase pair with modest cleaving efficiency.

As above, this reviewer considers that the joint use of Cas9n and VQR brings modest benefit, little methodological novelty and limited scientific significance. However, every reviewers/editors might have different opinions in this regard and, needless to say, the editors have the discretion in deciding whether the technological advance in this paper suffices for publication in *Communications Biology*.

2. Reproducibility of the one-step creation of mono-, bi-, and triallelic APP gene disruption:

This reviewer still believes that the authors were fortunate as they successfully generated all three different APP-disrupted alleles at a time. However, in the authors' rebuttal letter, they showed additional data replicating the one-step generation of the three genotypes with this procedure, suggesting its fair reproducibility. Accordingly, the authors are suggested incorporating these figures in the manuscript and referring to them in Results or Discussion to emphasize the reproducibility of the procedure, because readers of this paper may have the same concern with this reviewer's.

3. Strategy for genotyping of APP-disrupted clones:

The authors re-genotyped APP-disrupted cell clones using NGS-based deep sequencing. In addition, they performed array CGH analysis with some of the cell clones. These procedures should make their genotyping more convincing than Sanger-based analyses performed in their original manuscript. As a caveat, some clones delineated in new Fig. 1e and a figure in the rebuttal letter demonstrate near 50%–50% ratios of two different alleles, and are called as bi- or triallelic APP disruption. This is presumably due to large deletions in one of three APP alleles that were not amplified by PCR prior to NGS sequencing. However, the authors should try more to determine exact APP genotypes in these clones. A PCR amplicon for deep sequencing can be extended up to ~5 kb if needed to amplify the currently unamplified APP alleles (This reviewer assumes that the current PCR amplicon is much smaller than 5 kb, although no information regarding the current PCR amplicons for deep sequencing is provided in the manuscript). The clarification of large deletions may help exclude a possibility that the deletions involve entire APP exon 16 which results in transcriptional exon 16 skipping which the authors seem not to prefer. If the authors find redoing the NGS-based deep sequencing too laborious/costly, Sanger sequencing of the amplified APP alleles carrying large deletions may also be informative to prove the presence of such alleles with large deletions in addition to the other alleles already identified via NGS.

In Table S3, indications only by "Amp" and "Del" are unclear. Estimated copy numbers of the genomic regions should be shown. In addition, "full data availability statement" for NGS and CGH data is missing from the manuscript.

Re: Specific Comments in the first-round review

1., 2., 4.-6.

The authors sufficiently addressed requests/comments made by this reviewer in their revised manuscript.

3. Additional experiments supporting a role for the APP-PEG3-BAX axis:

The authors carried out PI staining of APP cells, Iso cells, and APP cells after drug treatments, in line with a recommendation from this reviewer, and confirmed their original data obtained with Caspase-3 staining. The authors also treated APP cells with a cytochrome C release blocker (CRI) and processed the cells for Caspase-3 and PI staining, confirming their original data obtained using a BAX inhibitor (BIP). However, the use of CRI and BIP in APP cells is insufficient to establish the contribution of the APP-PEG3-BAX axis to cellular apoptosis, because the role for PEG3 overexpression in mediating apoptotic signal to BAX in the context of APP duplication is not evidenced. According to data in this study, one can conclude that APP duplication induces PEG3 overexpression and BAX-mediated intrinsic apoptosis. As such, the authors should modify the last sentence in Results and two sentences in Discussion (lines 247-249 and 323-325). In the legend for Fig. S4, (e) and (f) should read (c) and (d), respectively.

Reviewer #2 (Remarks to the Author):

The authors have answered all the questions raised and provided additional information accordingly

Reviewer #3 (Remarks to the Author):

The authors attempted to address the previous claims and they did for a majority of the concerns. However, they did not provide clear responses to the following issues.

1. Despite the authors' claim that they drew technical advances in this study, I don't agree with that. Just combining two technology is not deemed new.

2. The authors provide very contradictory response for two claims. For comment 3, they responded that Cas9nVQR with sgRNA1 and sgRNA2 generated an editing efficiency of $40 \pm 3.5\%$ in iPSCs, which is tremendously high efficiency even for wild-type Cas9. However, for comment 10, they asserted that iPSCs is even difficult to transfect. If so, how could you derive editing efficiency of 40%?

Many of comments were addressed, while important questions remain still elusive.

Point-by-point Responses to Reviewers

COMMSBIO-20-0728A

Reviewer #1 (Remarks to the Author):

The authors extensively revised the manuscript supplementing several new experimental data. This reviewer finds significant improvement in the manuscript, nonetheless, has critiques and recommendations as listed below.

We thank the reviewer for commenting on the significant improvement of the revised manuscript. Our point-by-point responses to the reviewer's critiques and recommendations are detailed below.

Re: General Comments in the first-round review

1. Extent of methodological advance:

Although this reviewer does not disagree with the expanded genome targetability achieved by the combined use of Cas9 (D10A) and VQR mutation compared to the use of Cas9 (D10A) nickase pairs, in this reviewer's opinion, genome editing in this study was achieved simply via the joint use of a few pre-existing technologies and there is no notable synergistic effect made by their joint use. In addition, there is no surprise in the creation of 722-kb deletion when two pairs of complementary nicks are introduced at the same location on the tandem duplication of (probably) 722-kb DNA region. Ran et al. have already created genomic deletions using two Cas9 nickase pairs, although the deletions were smaller (~ 6 kb; Ran et al. Cell, 154: 1380–9, 2013). Furthermore, it is likely a common knowledge that both mono- and multiallelic knockouts can be generated when an inefficient Cas9 nuclease against an autosomal gene is introduced into cells. This study appears to simply reproduce the same result using a nickase pair with modest cleaving efficiency.

As above, this reviewer considers that the joint use of Cas9n and VQR brings modest benefit, little methodological novelty and limited scientific significance. However, every reviewers/editors might have different opinions in this regard and, needless to say, the editors have the discretion in deciding whether the technological advance in this paper suffices for publication in Communications Biology.

We apologize for not making our point clearer. We would like to clarify that the significance of this study is that it demonstrates the rapid, flexible, and efficient manipulation of gene dosage using CRISPR/Cas9 nickases in human iPSCs for disease modeling. The implications of developing this approach far exceed the joint use of Cas9n and VQR. To support this notion, we have performed an additional bioinformatics analysis and provide further explanations as follows:

1) To evaluate the synergistic effect of Cas9n and VQR, we performed an additional genome-wide coverage analysis for Cas9nQVR versus Cas9n based on the established

sgRNA design principle (Ran et al., *Cell* 154, 1380–1389, 2013; see below, **Supplementary Fig. S1a–d**). Cas9n can target 58.1% of the 3,209,286,105 sites in the human reference genome (GRCh38). In contrast, the Cas9nVQR developed in our study can increase the targetability to 79.8% when used alone and to 90.3% when used together with Cas9n; this represents 695,373,139 and 1,032,629,268 additional genomic sites, respectively. Furthermore, this study experimentally demonstrates that Cas9nVQR edits *APP* exon 16 sites that cannot be edited by Cas9n. Therefore, this study provides experimental evidence for a simple and efficient strategy for paired nickase-mediated genome editing that covers 90.3% of human genomic sites, which enables the study of expanded genetic variants associated with human diseases.

2) While Ran et al. demonstrated genomic deletions of ~6-kb using Cas9n (Ran et al., *Cell* 154, 1380–1389, 2013), we provide the first experimental evidence that Cas9nVQR, which we developed herein, can create 722-kb genomic deletions (more than 100 fold larger). Such large deletions enable deletion studies of large disease-associated genomic rearrangements and gene duplications, that is, in instances in which the entire *APP* duplication region is deleted. Therefore, our study vastly enhances the flexibility and robustness of Cas9nVQR.

3) While Cas9n exhibits enhanced specificity, it is assumed to be less efficient for genome editing than Cas9 nuclease. Our study serves as the first proof-of-concept that Cas9nVQR is highly efficient for genome editing in iPSCs ($40 \pm 3.5\%$ editing efficiency), enabling the simultaneous generation of isogenic iPSC lines with monoallelic, biallelic, or triallelic knockout of *APP* by screening fewer than 20 clones within 1 month. Therefore, our findings provide critical experimental evidence to support the adoption and application of Cas9nVQR in genome-editing studies for disease modeling.

We sincerely hope the reviewer agrees that our findings provide important knowledge for

genome-editing tool selection and support the use of Cas9 nickase over Cas9 nuclease when efficient and specific genome editing is required for disease modeling in human iPSCs. Accordingly, we have removed the claims regarding technological advances and incorporated the above points into the *Discussion*.

2. Reproducibility of the one-step creation of mono-, bi-, and triallelic APP gene disruption:

This reviewer still believes that the authors were fortunate as they successfully generated all three different APP-disrupted alleles at a time. However, in the authors' rebuttal letter, they showed additional data replicating the one-step generation of the three genotypes with this procedure, suggesting its fair reproducibility. Accordingly, the authors are suggested incorporating these figures in the manuscript and referring to them in Results or Discussion to emphasize the reproducibility of the procedure, because readers of this paper may have the same concern with this reviewer's.

As suggested by the reviewer, we have incorporated the data demonstrating the reproducibility of the one-step creation of *APP* gene disruption in the **Supplementary Fig. S3** and referred to them in the *Results*.

3. Strategy for genotyping of APP-disrupted clones:

The authors re-genotyped APP-disrupted cell clones using NGS-based deep sequencing. In addition, they performed array CGH analysis with some of the cell clones. These procedures should make their genotyping more convincing than Sanger-based analyses performed in their original manuscript. As a caveat, some clones delineated in new Fig. 1e and a figure in the rebuttal letter demonstrate near 50%–50% ratios of two different alleles, and are called as bi- or triallelic APP disruption. This is presumably due to large deletions in one of three APP alleles that were not amplified by PCR prior to NGS sequencing. However, the authors should try more to determine exact APP genotypes in these clones. A PCR amplicon for deep sequencing can be extended up to ~5 kb if needed to amplify the currently unamplified APP alleles (This reviewer assumes that the current PCR amplicon is much smaller than 5 kb, although no information regarding the current PCR amplicons for deep sequencing is provided in the manuscript). The clarification of large deletions may help exclude a possibility that the deletions involve entire APP exon 16 which results in transcriptional exon 16 skipping which the authors seem not to prefer. If the authors find redoing the NGS-based deep sequencing too laborious/costly, Sanger sequencing of the amplified APP alleles carrying large deletions may also be informative to prove the presence of such alleles with large deletions in addition to the other alleles already identified via NGS.

In Table S3, indications only by “Amp” and “Del” are unclear. Estimated copy numbers of the genomic regions should be shown. In addition, “full data availability statement” for NGS and CGH data is missing from the manuscript.

We thank the reviewer for raising this possibility. To demonstrate there are no large deletions, we conducted an additional analysis. We generated ~5.2-kb PCR amplicons surrounding *APP* exon 16 and detected no large deletion (i.e., >1-kb) by gel electrophoresis in the edited clones including those exhibiting near 1:1 ratios of two different alleles. We also generated ~1.2-kb

PCR amplicons surrounding *APP* exon 16 for Sanger sequencing, which detected no medium-sized deletions (i.e., >100-bp) in these clones. This “scaling down” genotyping strategy ensures there are no large- or medium-sized deletions. We have incorporated these data into the revised manuscript (**Supplementary Fig. S2**) and included the primer sequences for 5.2-kb PCR, 1.2-kb Sanger sequencing, and 150-bp deep sequencing in the *Methods*.

As the reviewer suggests, we have presented the estimated copy numbers of the genomic regions in **Supplementary Table S3** and provided information on NGS and CGH data under *Data availability* in the *Methods* accordingly.

Re: Specific Comments in the first-round review

1., 2., 4.–6.

The authors sufficiently addressed requests/comments made by this reviewer in their revised manuscript.

3. Additional experiments supporting a role for the APP-PEG3-BAX axis:

The authors carried out PI staining of APP cells, Iso cells, and APP cells after drug treatments, in line with a recommendation from this reviewer, and confirmed their original data obtained with Caspase-3 staining. The authors also treated APP cells with a cytochrome C release blocker (CRI) and processed the cells for Caspase-3 and PI staining, confirming their original data obtained using a BAX inhibitor (BIP). However, the use of CRI and BIP in APP cells is insufficient to establish the contribution of the APP-PEG3-BAX axis to cellular apoptosis, because the role for PEG3 overexpression in mediating apoptotic signal to BAX in the context of APP duplication is not evidenced. According to data in this study, one can conclude that APP duplication induces PEG3 overexpression and BAX-mediated intrinsic apoptosis. As such, the authors should modify the last sentence in Results and two sentences in Discussion (lanes 247–249 and 323–325).

In the legend for Fig. S4, (e) and (f) should read (c) and (d), respectively.

As suggested by the reviewer, we have revised the sentences in question in the *Results*, *Discussion*, and legend for **Fig. S4** as follows:

Results:

“These results collectively reveal that APP duplication induces PEG3 upregulation and BAX-mediated neuronal apoptosis in a human iPSC model of familial AD carrying *APP* duplication.”

Discussion:

“Specifically, we show that *APP* duplication induces PEG3 upregulation and BAX-mediated neuronal apoptosis.”

“Our study also demonstrates that *APP* regulates Tau expression and hyperphosphorylation and contributes to PEG3 upregulation and BAX-mediated

neuronal apoptosis in an isogenic background.”

Legend for Fig. S4:

“(c) Immunofluorescence images showing PI-positive cells in APP neurons at 28 days in vitro treated with vehicle control (Con), 100 μ M Bax inhibitor peptide V5 (BIP-V5), or 5 μ M cytochrome C release inhibitor (CRI) for 7 days. Scale bars: 50 μ m. (d) Percentage of PI-positive apoptotic cells among all neurons.”

Reviewer #2 (Remarks to the Author):

The authors have answered all the questions raised and provided additional information accordingly

Reviewer #3 (Remarks to the Author):

The authors attempted to address the previous claims and they did for a majority of the concerns. However, they did not provide clear responses to the following issues.

1. Despite the authors' claim that they drew technical advances in this study, I don't agree with that. Just combining two technology is not deemed new.

We apologize for not making our point clearer. We would like to clarify that the significance of this study is that it demonstrates significant improvements in the targetability, efficiency, and flexibility of CRISPR/Cas9 nickases to edit gene dosage in human iPSCs for disease modeling. To support this, we have performed an additional bioinformatics analysis and provide further explanations below:

(1) Targetability: We performed an additional genome-wide coverage analysis for Cas9nQVR versus Cas9n based on the established sgRNA design principle (Ran et al., *Cell* 154, 1380–1389, 2013; **see below, Supplementary Fig. S1a–d**). Cas9n can target 58.1% of the 3,209,286,105 sites in the human reference genome (GRCh38). In contrast, the Cas9nVQR developed in our study can increase the targetability to 79.8% when used alone and to 90.3% when used together with Cas9n; this represents 695,373,139 and 1,032,629,268 additional genomic sites, respectively. Furthermore, this study experimentally demonstrates that Cas9nVQR edits *APP* exon 16 sites that cannot be edited by Cas9n. Therefore, this study provides experimental evidence for a simple and efficient strategy for paired nickase-mediated genome editing that covers 90.3% of human genomic sites, which enables the study of expanded genetic variants associated with human diseases.

2) Efficiency: While Cas9n exhibits enhanced specificity, it is assumed to be less efficient for genome editing than Cas9 nuclease. Our study serves as the first proof-of-concept that Cas9nVQR is highly efficient for genome editing in iPSCs ($40 \pm 3.5\%$ editing efficiency), which enables the simultaneous generation of isogenic iPSC lines with monoallelic, biallelic, or triallelic knockout of *APP* by screening fewer than 20 clones within 1 month. Therefore, our findings provide critical experimental evidence supporting the adoption and application of Cas9nVQR in genome-editing studies for disease modeling.

3) Flexibility: Our study provides the first experimental evidence that Cas9nVQR, which we developed herein, can create 722-kb genomic deletions, which is more than 100 times larger than that reported previously (~ 6 kb; Ran et al. *Cell*, 154: 1380–1389, 2013). Such large deletions enable the study of large disease-associated genomic rearrangements and gene duplications—in this case, when we delete the entire *APP* duplication region. Therefore, our study vastly expands the flexibility and robustness of Cas9nVQR.

We sincerely hope the reviewer agrees that our findings provide important knowledge for genome-editing tool selection and support the use of Cas9 nickase over Cas9 nuclease when efficient and specific genome editing is required for disease modeling in human iPSCs. Accordingly, we have removed the claims regarding technical advances and incorporated these points into the *Discussion*.

2. The authors provide very contradictory response for two claims. For comment 3, they responded that Cas9nVQR with sgRNA1 and sgRNA2 generated an editing efficiency of $40 \pm 3.5\%$ in iPSCs, which is tremendously high efficiency even for wild-type Cas9. However, for comment 10, they asserted that iPSCs is even difficult to transfect. If so, how could you derive editing efficiency of 40%?

We thank the reviewer for commenting on the tremendously high efficiency of Cas9nVQR in our study and apologize for not making this point clearer. Editing efficiency is calculated by examining the percentage of edited reads among all reads in the Cas9nVQR-transfected cells. Regarding comment (3), an editing efficiency of $40 \pm 3.5\%$ was reported in the iPSCs after transfection and GFP sorting. The editing efficiency was unaffected by the low transfection efficiency of iPSCs ($<10\%$).

Many of comments were addressed, while important questions remain still elusive.

We sincerely hope that our point-to-point responses have adequately addressed the concerns raised by the reviewer.

REVIEWERS' COMMENTS:

Reviewer #1 (Remarks to the Author):

This reviewer has no more constructive comments/suggestions which may help improve the manuscript.

Reviewer #3 (Remarks to the Author):

The authors have endeavored to address the previous claims with clearer description. In particular, the procedure that the indel efficiency of more than 40% was obtained by FACS sorting was mentioned so as for readers not to be confused about the results.

Going back to the beginning, I have a some doubt that, if the high efficiency was obtained by FACS sorting, then what's new with this study in spite of the their assertions on targetability, efficiency, and flexibility.

Decision on this part would be at the editor's discretion.